# The distinction of CPR bacteria from other bacteria based on protein family content

Raphaël Méheust [1,2], David Burstein [1,3,8], Cindy J. Castelle[1,2,4] & Jillian F. Banfield [1,2,4,5,6,7]

Candidate phyla radiation (CPR) bacteria separate phylogenetically from other bacteria, but the organismal distribution of their protein families remains unclear. Here, we leveraged sequences from thousands of uncultivated organisms and identified protein families that co-occur in genomes, thus are likely foundational for lineage capacities. Protein family presence/absence patterns cluster CPR bacteria together, and away from all other bacteria and archaea, partly due to proteins without recognizable homology to proteins in other bacteria. Some are likely involved in cell-cell interactions and potentially important for episymbiotic lifestyles. The diversity of protein family combinations in CPR may exceed that of all other bacteria. Over the bacterial tree, protein family presence/absence patterns broadly recapitulate phylogenetic structure, suggesting persistence of core sets of proteins since lineage divergence. The CPR could have arisen in an episode of dramatic but heterogeneous genome reduction or from a protogenote community and co-evolved with other bacteria.

[1] Department of Earth and Planetary Science, University of California, Berkeley, Berkeley, CA 94720, USA. [2] Innovative Genomics Institute, Berkeley, CA 94704, USA. [3] California Institute for Quantitative Biosciences (QB3), University of California, Berkeley, Berkeley, CA 94720, USA. [4] Chan Zuckerberg Biohub, San Francisco, CA 94158, USA. [5] University of Melbourne, Melbourne, VIC 3010, Australia. [6] Lawrence Berkeley National Laboratory, Berkeley, CA 94720, USA. [7] Department of Environmental Science, Policy and Management, University of California, Berkeley, Berkeley, CA 94720, USA. [8] Present address: School of Molecular and Cell Biology and Biotechnology, George S. Wise Faculty of Life Sciences, Tel Aviv University, Tel Aviv 69978, Israel. Correspondence and requests for materials should be addressed to J.F.B. (email: jbanfield@berkeley.edu)

Metagenomic investigations of microbial communities have generated genomes for a huge diversity of bacteria and archaea, many from little studied or previously unknown phyla[1]. For example, a study of an aquifer near the town of Rifle, Colorado generated 49 draft genomes for several groups of bacteria, some of which were previously known only based on 16S rRNA gene surveys and others that were previously unknown[2]. Draft genomes for bacteria from related lineages were obtained in a single cell sequencing study that targeted samples from a broader variety of environment types[3]. Based on the consistently small predicted genome sizes for bacteria from these groups, groundwater filtration experiments targeting ultra-small organisms were conducted to provide cells for imaging[4] and DNA for increased genomic sampling. The approach yielded almost 800 genomes from a remarkable variety of lineages that were placed together phylogenetically. This monophyletic group was described as the candidate phyla radiation (CPR)[5]. CPR bacterial genomes have since been recovered from the human microbiome[6], drinking water[7], marine sediment[8], deep subsurface sediments[9], soil[10], the dolphin mouth[11] and other environments[12]. Thus, it appears that CPR bacteria are both hugely diverse and widespread across earth's environments.

Metabolic analyses of CPR genomes consistently highlight major deficits in biosynthetic potential, leading to the prediction that most of these bacteria live as symbionts. Cultivation from human oral samples highlighted the attachment of a CPR member of the lineage Saccharibacteria (TM7) to the surface of an *Actinomyces odontolyticus* bacteria[6]. Another episymbiotic association has been described between a CPR organism from the Parcubacteria superphylum and an eukaryotic host[13]. However, most CPR organisms are likely symbionts of bacteria or archaea, given their abundance and diversity in samples that have few, if any, eukaryotes[1].

When first described, the CPR was suggested to comprise at least 15% of all bacteria[5]. Subsequently, Hug et al.[14] placed a larger group of CPR genome sequences in context via construction of a three-domain tree and noted that the CPR could comprise as much as 50% of all bacterial diversity. The CPR was placed as the basal group in the bacterial domain in a concatenated ribosomal protein tree, but the deep branch positions were not sufficiently well supported to enable a conclusion regarding the point of divergence of CPR from other bacteria. The scale of the CPR is also controversial. For example, Parks et al.[15] suggested that the group comprises no more than 26.3% of bacterial phylum-level lineages.

To date, most studies have predicted CPR metabolic traits using one or a few genomes. Lacking are studies that look radiation-wide at the distribution of capacities that are widespread and thus likely contribute core functions, including those encoded by hypothetical proteins. Moreover, examination of genetic potential across the CPR and general comparisons of CPR and non-CPR bacteria have been very limited. Here, we leverage a large set of publicly available, good-quality genomes of CPR and non-CPR bacteria to address these questions. We clustered protein sequences from 3598 genomes into families and evaluated the distribution of these protein families over genomes. Given the large extent of divergence over the history of the bacterial domain and difficulties with accurately distinguishing orthologs from homologs, our approach considers homologous protein families[16]. By focusing only on protein families that are common in CPR bacteria and/or non-CPR bacteria, we demonstrate a major subdivision within the bacterial domain without reliance on gene or protein sequence phylogenies. The separation is, in part, due to proteins missing in CPR. However, we also characterize a set of 106 proteins ubiquitous in CPR while less abundant in other bacteria. Some of these are likely involved in cell–cell interactions

and potentially important for episymbiotic lifestyles. The diversity of combinations of protein families in CPR may exceed that of all other bacteria. Based on the present results and results from previous studies, we propose a scenario where the CPR have arisen from a protogenote community and co-evolved with other bacteria.

## Results

**Clustering of proteins and assessment of cluster quality.** We collected 3598 genomes from four published datasets[5,9,17,18]. The dataset includes 2321 CPR genomes from 65 distinct phyla (1,953,651 proteins), 1198 non-CPR bacterial genomes from 50 distinct phyla (3,018,597 proteins) and 79 archaeal genomes (89,709 proteins) (Fig. 1). Note that this huge sampling of Candidate Phyla was only possible due to genomes reconstructed in the last few years (Fig. 1). We clustered the 5,061,957 protein sequences in a two-step procedure (see Methods and Supplementary Fig. 1) to generate groups of homologous proteins. The objective was to convert amino acid sequences into units of a common language, allowing us to compare the proteomes across a huge diversity of genomes. This resulted in 22,977 clusters (representing 4,449,296 sequences) that were present in at least five distinct non-redundant and draft-quality genomes. These clusters are henceforth referred to as protein families.

To assess the extent to which the protein clusters group together proteins with shared functions, we analyzed some families with well-known functions, such as the 16 ribosomal proteins that are commonly used in phylogeny[19]. Because these proteins are highly conserved, we expect one protein family per ribosomal subunit. For instance, we expected to have all proteins annotated as the large subunit 3 (RPL3) be clustered into the same family. For 15 out 16 subunits, all proteins clustered into one single family (Supplementary Table 1). Only the large ribosomal subunit 2 clustered into two large families (fam004931 and fam006844). Close inspection showed that the two families were not merged because their corresponding HMMs matched only partly (based on the thresholds used, Supplementary Fig. 3A). The ribosomal proteins are among the slowest-evolving proteins, so one may expect that they easily cluster together. In order to assess the quality of the protein clustering on faster-evolving proteins, we performed the same analysis on non-ribosomal proteins. We annotated our protein dataset using the KEGG annotations[20] and systematically verified that the protein family groupings approximate functional annotations. The KEGG annotations in our dataset encompass 7700 unique annotations of various biological processes, including the fast-evolving defense mechanisms. For each of these 7700 annotations, we reported the family that contains the highest percentage of protein members annotated with that KEGG annotation. Most clusters were of good quality. For 89.2% of the annotations (6872 out of 7700) one family always contained >80% of the proteins (Supplementary Fig. 3B).

For each protein family with a KEGG annotation, we assessed the contamination of the protein family by computing the percentage of the proteins with KEGG annotations that differ from the dominant annotation (percentage annotation admixture). Most of the families contain only proteins with the same annotation, and 3608 families (78.7%) have <20% annotation admixture (Supplementary Fig. 3C). Although this metric is useful, we note that it is imperfect because two homologous proteins can have different KEGG annotations and thus cluster into the same protein family, increasing the apparent percentage of annotation admixture. For instance, the phenylalanyl-tRNA synthetase possesses two subunits alpha (K01889) and beta (K01890) that are homologous[21] and thus their protein sequences

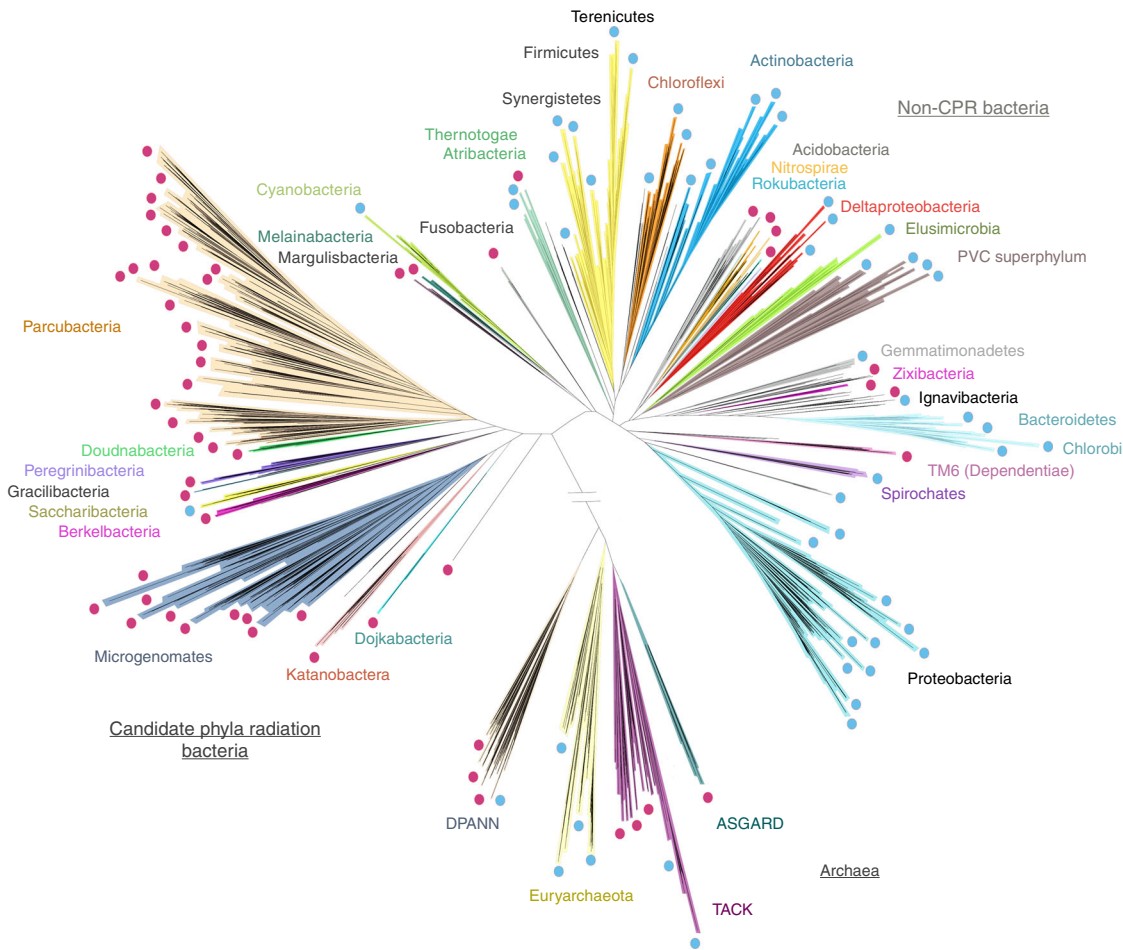

**Fig. 1** Schematic tree illustrating the phylogenetic sampling used in this study. Lineages that were included in the datasets are highlighted with a dot. Lineages lacking an isolated representative are highlighted with red dots. The number of genomes per lineages is available in Supplementary Fig. 2. The diagram is based on a tree published recently in ref. [1]

clustered into the same protein family (fam000662). This family shows a high annotation admixture of 45% because nearly half of their proteins members are annotated as alpha subunits (K01889) and half as beta subunits (K01890).

Although we used sensitive sequence-comparison methods and assessed the quality of the protein clustering, we cannot completely rule out the possibility that our pipeline failed to retrieve distant homology for highly divergent proteins. Small proteins and fast-evolving proteins are more likely to be affected[22]. This lack of sensitivity would result on the separation of homologous proteins into distinct families and would affect the results.

**Widespread proteins subdivide CPR from all other bacteria.** For definition of protein families, we chose a dataset that includes sequences from a huge diversity of uncultivated lineages and (unlike most reference genome datasets), genomes from the majority of all bacterial phyla (Fig. 1). We constructed an array of the 2890 non-redundant and draft-quality genomes (rows) vs. 22,977 protein families (columns) and hierarchically clustered the genomes based on profiles of protein family presence/absence. The families were also hierarchically clustered based on profiles of genome presence/absence (Fig. 2a). The distinct pattern of protein family presence/absence in CPR genomes separates them from almost all non-CPR bacteria and from archaea regardless

the methods of agglomerative hierarchical clustering used (Fig. 2a and Supplementary Fig. 4).

Most protein families cluster together due to co-existence in multiple genomes (blocks of black and orange dots in Fig. 2a). Strikingly, some blocks with numerous families are widespread in non-CPR bacteria while mostly absent in CPR (Fig. 2a), which may explain the observed separation of the CPR from the non-CPR bacteria.

We identified co-occurring blocks of protein families (sharing similar patterns of presence/absence across the genomes) using the Louvain algorithm[23]. We defined modules as blocks of co-occurring protein families containing at least 10 families. In all 15,137 protein families could be assigned to 236 modules. The remaining 7840 protein families were not assigned to a module with >10 families, and thus were not considered further. As the majority of protein families are fairly lineage specific (232 of the 236 modules are sparsely distributed among the 115 bacterial phyla; blue dots in Fig. 2b), they were excluded from further analysis so that we could focus on families that are widespread (orange dots in Fig. 2b). Ultimately, we analyzed four modules comprising 921 families and over three million protein sequences. Some of these modules also occur in archaeal genomes, so archaeal genomes were retained in the study.

Given their widespread distribution it is unsurprising that most the 921 families are involved in well-known functions, including replication, transcription and translation, basic metabolism

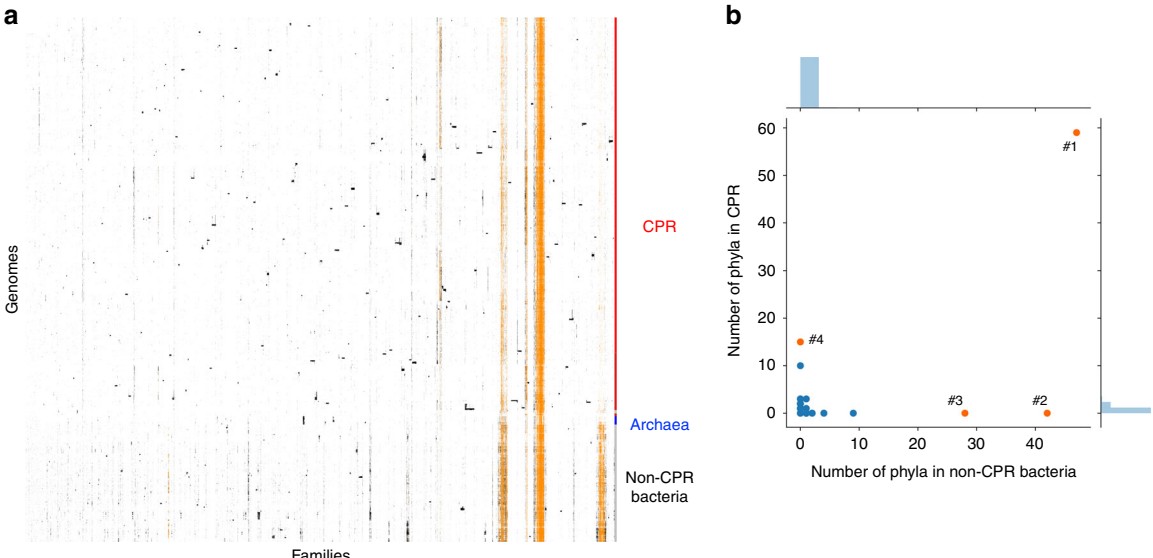

**Fig. 2** The distribution of the 22,977 families across the 2890 genomes. **a** The distribution of the 22,977 families (columns) across the 2890 prokaryotic genomes (rows). Data are clustered based on the presence (black)/absence (white) profiles (Jaccard distance, hierarchical clustering using a complete linkage). Archaea: blue, CPR: red, non-CPR bacteria: gray. The patterns in orange correspond to the presence/absence patterns of 921 widespread families. **b** The phyla distribution of the 235 modules of proteins in CPR (y axis) and non-CPR bacteria (x axis). Each dot corresponds to a module. The orange dots correspond to the four widespread modules on which further analyses focus

(energy, nucleotides, amino acids, cofactors and vitamins) and environmental interactions (membrane transport such as the Sec pathway) (Supplementary Data 1). One of the four modules is essentially ubiquitous across the dataset (orange dot #1 in Fig. 2b), two are present in at least 10 non-CPR bacterial phyla (orange dots #2 and 3 in Fig. 2b) but are mostly absent in CPR bacteria, and the fourth occurs in more than 10 CPR bacterial phyla (orange dot #4 in Fig. 2b).

We conducted an unsupervised clustering of the genomes based on the presence/absence profiles of the 921 families. Using the complete-linkage method to create the hierarchical clustering, the results clearly distinguish the CPR bacteria from other bacteria and archaea (Fig. 3a, b), as seen in Fig. 2. Using different methods of agglomerative hierarchical clustering, the separation is still apparent (Supplementary Fig. 5) and the genome clusterings obtained with the profiles of the 22,977 families (Supplementary Fig. 4) and with the 921 widely distributed families (Supplementary Fig. 5) are correlated (complete-linkage: 0.88, average-linkage: 0.92, single-linkage: 0.84, Supplementary Fig. 6). Thus, the 921 families may alone explain the separation between the CPR and the non-CPR bacteria. The interesting exception in the separation is the Dependentiae phylum (TM6), which is nested in the CPR group between Microgenomates and Parcubacteria (Fig. 3b) although phylogenetic trees based on core genes clearly place Dependentiae outside of the CPR[5]. Their nesting within the CPR occurs with both average-linkage and complete-linkage methods (although the Dependentiae phylum is more basal in the clusterings based on average linkage; Supplementary Figs. 4 and 5). Dependentiae were placed with the non-CPR bacteria only in clusterings based on single linkage. However, of all clustering methods, this showed the poorest congruence between phylogenetic and protein family trees (Supplementary Figs. 4–6). We discuss the placement of the Dependentiae phylum further below from the perspective of their gene content.

When the hierarchical clustering pattern from the y axis of Fig. 3a is rendered in a radial tree format (Fig. 3b) the correspondence between clusters based on the distribution of core protein families and phylogeny (Fig. 3c) is apparent

consistent with previous studies[24,25]. Based on complete-linkage, the cophenetic correlation between a maximum-likelihood phylogenetic tree is 0.70, based on average-linkage 0.67, and based on single-linkage, 0.54 (Supplementary Fig. 6). Within the CPR, clustering of genomes based on the protein family distribution patterns (Fig. 3b) is generally consistent with their clustering in the 14-ribosomal-protein phylogeny (Fig. 3c). The Microgenomates and Parcubacteria superphyla form two distinct groups, with the exception of Woykebacteria, which is expected to be within the Microgenomates (Fig. 3b). Doudnabacteria, Berkelbacteria, Kazan and the Peregrinibacteria are sibling to, or nested in, Parcubacteria, but it should be noted that ordering of deep branches is difficult using gene phylogenies. Saccharibacteria clusters with Dojkabacteria, Katanobacteria, and Woykebacteria, although it is normally sibling to the Parcubacteria superphylum.

The analysis was made using draft-quality genomes (>70% completeness). It is expected that such content-based analyses are affected by genome completeness. We analyzed if using high-quality genomes improved the congruence between the phylogenetic tree and the families-content tree. We re-analyzed the data using only high-quality genomes (1966 genomes with >90% completeness)[26] (Supplementary Fig. 7). Based on complete linkage, the cophenetic correlation between a maximum-likelihood phylogenetic tree is 0.70, based on average linkage 0.74, and based on single linkage, 0.67. Of note, the Dependentiae phylum is still nested within the CPR (Supplementary Fig. 7). The results are similar to those obtained using the >70% complete genomes (Fig. 3).

The analysis present in Figs. 2 and 3 used a genomic dataset that was notably enriched in CPR bacteria. To test whether the clear separation of CPR and non-CPR bacteria is an artifact of the choice of genomes, we created a second dataset of 2729 of publicly available NCBI genomes sampled approximately at the level of one per genus (see Methods). Out of 22,977 protein families, 15,305 were identified in this dataset and arrayed using the same approach as in Fig. 2a (Supplementary Fig. 8). The 921 widespread protein families were arrayed using the same approach as in Fig. 3a (Fig. 4a). As observed with the first dataset,

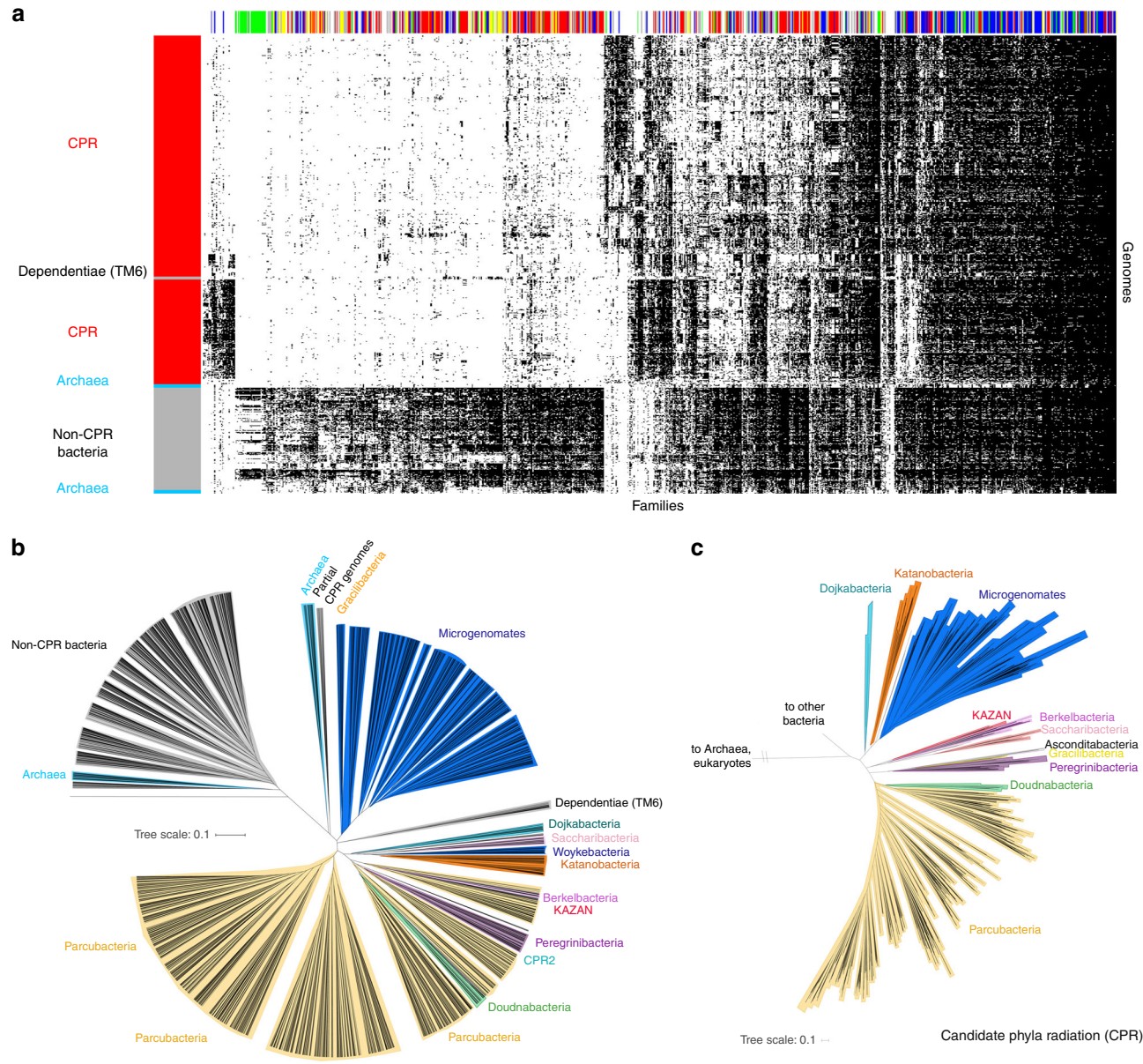

**Fig. 3** The distribution of 921 widely distributed protein families across the 2890 genomes. **a** The distribution of 921 widely distributed protein families (columns) in 2890 genomes (rows) from CPR bacteria (red), non-CPR bacteria (gray), and a few archaea (light blue) in a reference set with extensive sampling of genomes from metagenomes (thus including sequences from many candidate phyla). Data are clustered based on the presence (black)/ absence (white) profiles (Jaccard distance, complete linkage). Only draft-quality and non-redundant genomes were used. The colored top bar corresponds to the functional category of families (Metabolism: red, Genetic Information Processing: blue, Cellular Processes: green, Environmental Information Processing: yellow, Organismal systems: orange, Unclassified: gray, Unknown: white). **b** Tree resulting from the hierarchical clustering of the genomes based on the distributions of proteins families in panel **a**. **c** A phylogenetic tree of the CPR genomes present in the dataset. The maximum-likelihood tree was calculated based on the concatenation of 14 ribosomal proteins (L2, L3, L4, L5, L6, L14, L15, L18, L22, L24, S3, S8, S17, and S19) using the PROTCATLG model

the genome clusterings obtained with the profiles of the 15,305 families and with the 921 widely distributed families are correlated (Supplementary Fig. 8). The diagrams clearly separate CPR from non-CPR bacteria and from archaea, except in the diagram based on the profiles of the 921 families and the single-linkage clustering. Thus, we conclude that the major subdivision within the first dataset was not due to our choice of genomes or the environments they came from. Importantly, this NCBI genome dataset includes many genomes from symbionts with reduced genomes[27]. In no case could these genomes be placed within the CPR although several highly reduced genomes are

basal to the CPR group (discussed further in the manuscript with regard to their gene content as for TM6).

From the hierarchical clustering of the genomes in Fig. 4 we generated a tree representation analogous to that in Fig. 3b (Fig. 4b). Again, the correspondence between genome clusters based on protein family distribution and phylogeny is striking although several phyla are split into several groups. This is particularly apparent for highly diverse phyla such as Actino-bacteria, Firmicutes and Proteobacteria (Fig. 4b). These incon-gruences are due to differences in the sets of families (Fig. 4a). Interestingly, the groups are not correlated with particular

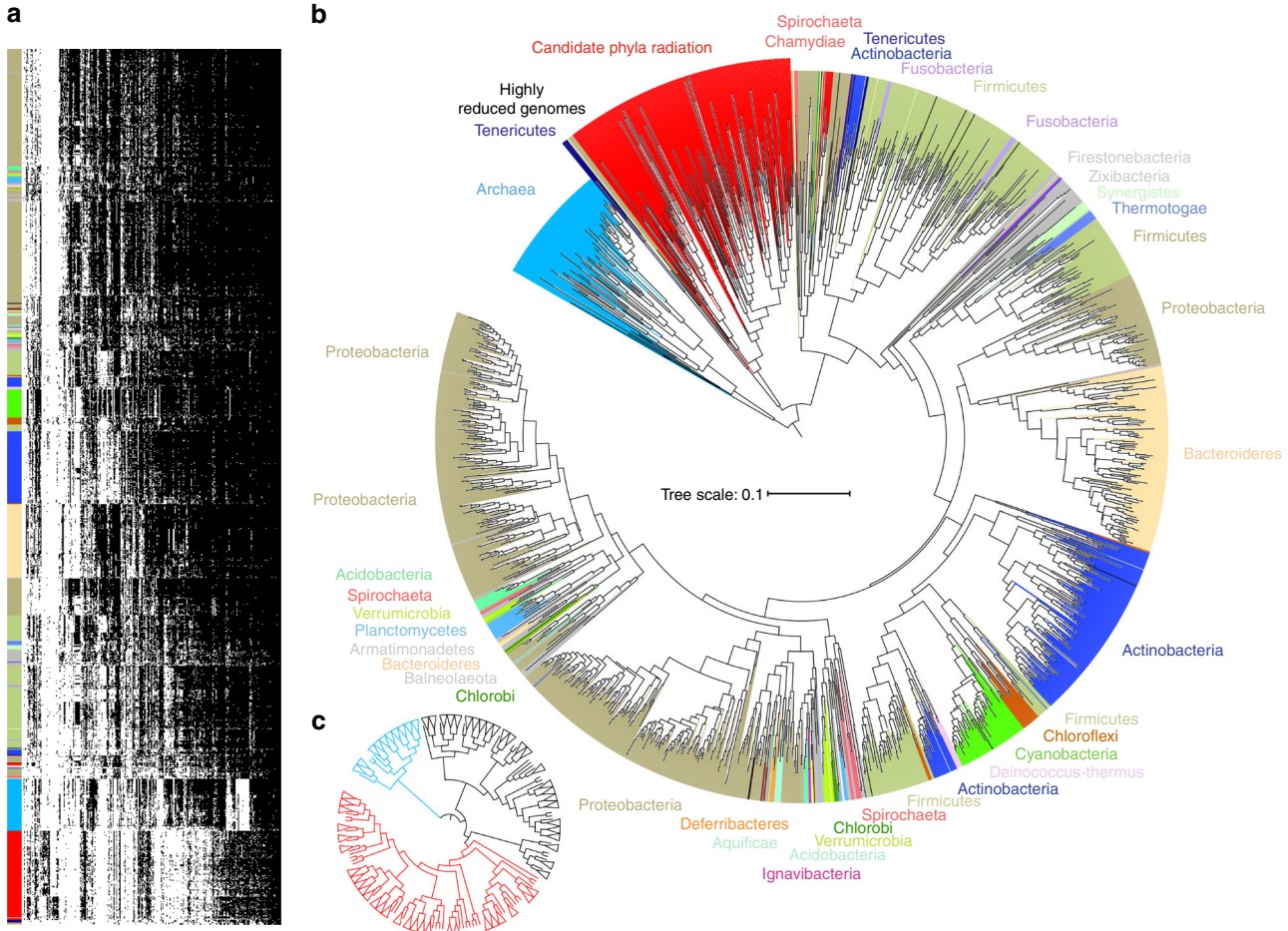

**Fig. 4** The distribution of protein families across representative genomes of the prokaryotic tree of life. **a** The distribution of 921 widely distributed protein families (columns) in 2616 draft-quality and non-redundant genomes (rows) from a reference set with extensive sampling of genomes from non-CPR bacteria. Genomes are clustered based on the presence (black)/absence (white) profiles (Jaccard distance, complete linkage). The order of the families is the same as in Fig. 3a. **b** Tree resulting from the hierarchical clustering of the genomes based on the distributions of proteins families in panel a. **c** The same tree with a collapsing of all branches that represented <25% of the maximum branch length (CPR are in red, Archaea in blue and non-CPR bacteria in gray)

subclades of Actinobacteria, Firmicutes or Proteobacteria. Other phyla, such as Cyanobacteria (bright green in Fig. 4a), have more consistent patterns of presence/absence of core protein families. This is reflected in the comparatively short branch lengths in Fig. 4b. In contrast, the branch lengths associated with the CPR bacteria are very long.

To evaluate branch length patterns through Fig. 4b, we collapsed all branches that represented less than 0.25 of the maximum branch length (Fig. 4c). In this rendering, 84 out of 85 of the Cyanobacteria collapse into a single wedge. The CPR bacteria comprise 89 wedges, non-CPR bacteria 71 wedges and Archaea, 24 wedges. Notably, DPANN archaea cluster separately from other archaea, consistent with their phylogenetic separation in some analyses[28]. The high representation of wedges of CPR relative to non-CPR bacteria is striking, given that CPR genomes represent only 11% of all genomes used in this analysis. Similar results were found when comparing the distributions of the similarities of the protein sets within the CPR and within the non-CPR bacteria (Supplementary Fig. 9) (Mann–Whitney–Wilcoxon test, $P = 0.0$). We attribute these results to high diversity in the subsets of core protein families present in genomes of organisms from across the CPR.

To test whether the protein clustering cutoffs strongly affected our results we performed another protein clustering without using cut-offs set during the HMM–HMM comparison (see Methods). We retrieved 1216 protein clusters that correspond to

the 921 widespread families. Using this distinct clustering, the CPR still separate from non-CPR bacteria and archaea in analyses that used both genome datasets (Supplementary Figs. 10 and 11). Thus, we conclude that, our results are robust regarding both genome selection (as tested using the NCBI genome dataset) and the protein clustering parameters.

**Biological capacities explain the singularity of CPR**. To explore the reasons for the genetic distinction of CPR from non-CPR bacteria and archaea we divided the 921 protein families into three sets based on their abundances in CPR and in non-CPR bacteria (Supplementary Fig. 12A). A set of 233 families are equally distributed across the bacteria and 688 families are either depleted or enriched in CPR bacteria. The set equally distributed in CPR and non-CPR bacteria contains families mostly involved in informational processes, primarily in translation (Supplementary Fig. 12A).

Of the 688 families, 582 families are rare in CPR yet very common in other bacteria (Fig. 3a and Supplementary Fig. 12A). As expected based on prior work, this set is enriched in families involved in metabolism (Fig. 3a and Supplementary Fig. 12A). Although the CPR bacteria are distinct from non-CPR bacteria due to their sparse metabolism and the presence of CPR-specific genes, they are not monolithic in terms of their metabolism[12]

(Supplementary Fig. 12B). For example, the genomes of the Peregrinibacteria encode far more metabolic families than Shapirobacteria (Microgenomates), Dojkabacteria and Gracilibacteria. Despite the comparatively high metabolic gene inventory of the Peregrinibacteria, they have far fewer capacities than, for example, Alphaproteobacteria (Supplementary Fig. 12B).

The set also contains families involved in informational systems, including RNA polymerase σ54 sigma factor (fam001584), the GTPase Der (fam002205) and ribosomal protein L30 (fam003736)[5]. The 16S rRNA processing protein RimM (fam001806) is also missing in most CPR genomes and the ribosome-binding factor A (RbfA, fam000917) was found in 969 out of 2177 CPR genomes (Fig. 5). In *Escherichia coli*, mutants lacking RimM or RbfA showed a reduced efficiency in the processing of the 16S rRNA[29]. RimP (fam001726) is also essentially missing in CPR, and is important for the maturation of the 30S ribosomal subunits in *E. coli*[30]. Almost all bacteria and Eukarya have RsfS/RfsA (fam001828), which encodes a ribosomal silencing factor[31], but this is absent in CPR. In *E. coli*, RsfS/RfsA interacts with ribosomal protein RPL14 and inhibits translation when nutrients are depleted[32] (Fig. 5).

Interestingly, 44 depleted CPR families are annotated as involved in membrane transport, many of them are almost absent in CPR while widespread in non-CPR bacteria (Supplementary Data 1). One of these families is the fluoride exporter Fluc family (also named crcB, fam003329)[33]. Fluoride is pervasive in the environment and is toxic for single-cell organisms as it inhibits two enzymes involved in nucleotide biosynthesis and glycolysis[34]. The Fluc family has been described as widespread in the bacteria but is almost absent in CPR (detected in one CPR genome)[33]. The fluoride exporter is not the only transporter involved in detoxification, few CPR genomes encode the cation diffusion facilitator family (fam000411, detected in 320 CPR genomes), the arsenical resistance-3 (acr3) family (fam000792, detected in 58 CPR genomes) or the chromate transporter (fam001093, detected in one CPR genome). This raises the possibility of the existence of an alternative system to remove these toxic ions. Several uptake systems are also depleted. The potassium uptake system (trk system) found in a large number of bacterial species is also almost completely absent in CPR (fam000602 and fam001673, detected in 28 and 7 genomes, respectively) as well as the ammonium transporter Amt (fam001162), the ferrous transport Feo system (fam000582 and fam001321), and the CorA magnesium transporter family (fam000804, detected in CPR 919 genomes). The three inner membrane proteins TonB-ExbB-ExbD (fam000056, fam000382, and fam000368) are also missing in CPR. The complex interacts with the outer membrane proteins that bind and transport siderophores as well as vitamin B12, nickel chelates, and carbohydrates in Gram-negative bacteria[35]. This observation is consistent with the proposed absence of an outer membrane in CPR bacteria.

Finally, 106 protein families are enriched in CPR, rare in non-CPR bacteria, and are discussed in detail in the next session. Importantly, when these 106 protein families are removed from the set of widespread families and the analysis re-performed, the CPR bacteria are still different from all other bacteria regarding both genome datasets (Supplementary Figs. 13 and 14).

**CPR families are linked to pili and cell–cell interactions**. As noted above, 106 protein families are enriched in CPR relative to non-CPR bacteria (Fig. 6 and Supplementary Data 1). The majority of the 106 families have poor functional annotations (Supplementary Fig. 12A and Supplementary Data 1). However, 76 families are comprised of proteins with at least one predicted

transmembrane helix (Supplementary Data 1), and many are predicted to have membrane or extracellular localizations (Supplementary Fig. 12C and Supplementary Data 1). Eight have more than four transmembrane helices, and may be involved in transport (Supplementary Data 1). For instance, the family fam001364 has five transmembrane helices predicted and is annotated as a mechanosensitive ion channel according to the transporter database TCDB[36] (Supplementary Data 1).

Interestingly, 51 of the 106 protein families are widespread in all CPR bacteria; the others are enriched in either Microgenomates (35 families, center, left side of Fig. 6) or Parcubacteria (20 families, center top in Fig. 6). Those associated with Microgenomates or Parcubacteria are primarily hypothetical proteins. However, 14 of 35 protein families enriched in Microgenomates and 12 of 20 protein families enriched in Parcubacteria are predicted to be localized in the membrane (Supplementary Data 1).

Given that most CPR bacteria lack the ability to de novo synthesize nucleotides, it is anticipated that their cells scavenge DNA[37]. The DNA processing protein A (DprA) and protein competence protein EC (ComEC) are essential components of the DNA uptake machinery[38,39]. The DprA component (fam000839) is widespread and equally distributed in both CPR and non-CPR bacteria (Supplementary Data 1). We identified one family (fam000603) annotated as ComEC in 51% of genomes of the non-CPR bacteria, but it is more abundant in the CPR bacteria (detected in 86% of the CPR genomes, Supplementary Data 1). Interestingly, 29% of the genomes of non-CPR bacteria have a different version of ComEC fused with a metallo-beta-lactamase domain, and are thus clustered into a distinct family that is related to the metallo-beta-lactamase (fam000058). This protein fusion is essentially absent in CPR bacteria (found in only six CPR genomes). Two other components, ComFC/comFA (fam000096) and ComEA (fam000152), are present, although they are not essential to the natural transformation machinery[38]. ComFC/comFA is slightly depleted in CPR based on statistical testing, although the protein family is quite widespread in the CPR genomes (detected in 2068 genomes). ComEA is involved in DNA binding and clusters with the protein sequences homologous to the RuvA domain (fam000152). The ComEA KEGG annotation (K02237; one of the two KEGG annotations in fam000152) is detected in only 31% CPR genomes and in 67% non-CPR bacterial genomes, suggesting that some CPR genoms may possess an alternative mechanism for DNA binding (Fig. 5).

In competent bacteria, a correlation has been shown between the ability to take up exogenous DNA and the presence of pili on the cell surface[40,41]. We found that one family (fam000005) enriched and widespread in CPR is a cluster of pilin proteins, the subunits of pili. These typically have a single-transmembrane domain in their first 50 amino -acids[42]. These pilin proteins are part a type IV pili (T4P) system that includes other components that are enriched in the genomes of CPR bacteria but are also present in non-CPR bacteria[43] (Fig. 6). These components comprise both the ATPase assembly PilB and the ATPase twitching motility PilT (both present in fam000300), the three membrane platform components PilC, PilO and PilN (fam000383, fam000067, and fam000103), and, finally, the GspL domain PilM (fam000148). The prepilin peptidase PilD (fam000587) is evenly distributed and widespread in bacteria and in CPR. All of these components co-localize in numerous CPR genomes. Importantly, we did not find the PilQ component, which is required to extrude the pilus filament across the outer membrane of gram-negative bacteria[39] (Fig. 5), consistent with the observations from microscopy that suggest CPR do not have a gram negative cell envelope[4].

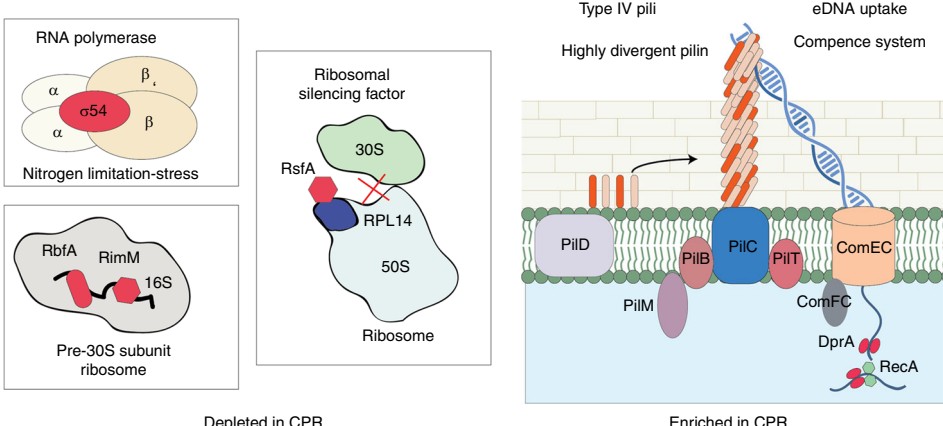

**Fig. 5** Example of proteins that are depleted or enriched in CPR. In the left panel are represented four proteins (colored in red) that are involved in informational machineries and are depleted in CPR, yet widespread and important in non-CPR bacteria. In the right panel is represented a schematic model for the type IV pili and the competence systems that appear widespread in CPR bacteria

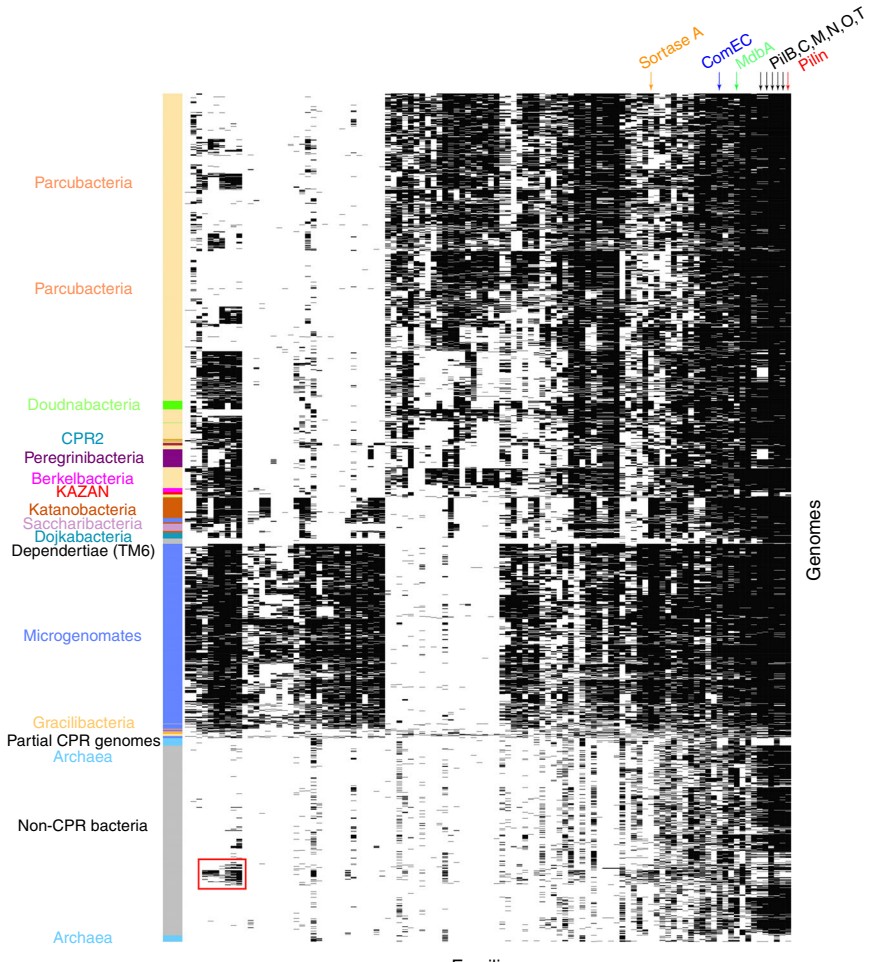

**Fig. 6** Distribution of the 106 families that are enriched in CPR relative to non-CPR bacteria. The distribution of 106 protein families enriched in CPR (columns) in 2890 draft-quality and non-redundant genomes (rows) from a reference set with extensive sampling of genomes from metagenomes. The order of the families and the genomes is the same as in Fig. 3A

Full-length type IV pilin precursors are secreted by the Sec pathway in unfolded states in gram-positive bacteria[42]. A thiol-disulfide oxidoreductase (fam000318) is one of the protein families enriched in CPR bacteria (Fig. 6) and may be involved in ensuring correct folding of the pilins. These proteins show

similarity to membrane-bound oxidoreductase MdbA, which is found in the gram-positive *Actinomyces oris*[44] and *Corynebacterium diphtheriae*[45]. In these organisms, MdbA catalyzes disulfide bond formation in secreted proteins, a reaction that is important for protein stability and function[46]. In *Actinomyces oris*, one of

these secreted proteins is the FimA pilin[44]. Similarly to MdbA, 45% of the proteins from the family fam000318 are predicted to be anchored in the cell wall (Supplementary Data 1) and the catalytic CxxC motif required for disulfide bond formation is conserved in 89% CPR proteins. Interestingly, the third family of proteins most frequently found in CPR genomes adjacent to proteins of fam000318 is the Vitamin K epoxide reductase (fam000898, VKOR, adjacent in 88 out of 750 CPR genomes that have both families). VKOR re-oxidizes MdbA in *Actinomyces oris*[47].

More than half of all CPR have fam000526, a sortase, whereas this is detected in less than 10% of non-CPR bacteria in the first dataset and 30% in the more taxonomically balanced second dataset (Supplementary Data 1). The family is near ubiquitous in Microgenomates but patchily detected in other CPR groups (Fig. 6). The low incidence in non-CPR is expected, given the association of this function with gram-positive bacteria (Firmicutes and Actinobacteria, as well as Chloroflexi). Seven other families are enriched in Gram-positive bacteria but also in Cyanobacteria and Chloroflexi, where they define a conserved pattern of presence/absence (Fig. 6, the red box). The family fam000574 contains a domain of unknown function (DUF4012). However, a gene annotated as DUF4012 has been localized in the capsular polysaccharides/exopolysaccharides gene cluster in the gram-positive bacteria *Bifidobacterium longum* 105A[48]. Consequently, the gene has been proposed to encode an auxiliary protein for envelope protein[48]. Interestingly, another family, fam000680, is annotated as a putative regulatory protein involved in exocellular polysaccharide biosynthesis (Supplementary Data 1). Capsular polysaccharides/exopolysaccharides are thought to be critical in host–microbe interactions[48]. Family fam000706 seems also to be involved in cell envelope function as it contains a putative peptidoglycan binding domain and the vancomycin resistance domain W (vanW). Glycopeptide antibiotic vancomycin inhibits the extracellular steps of bacterial peptidoglycan synthesis. Although the function of vanW is unknown, it has been found in the VanB-type glycopeptide resistance gene cluster in the Gram-positive *Enterococcus faecalis* V583. These observations strengthen the prediction that the cell envelope of CPR bacteria is likely more similar to that of gram-positive compared to Gram-negative bacteria. The remaining families fam000442, fam000682, fam001505 and fam000479 have no predicted annotations. Given that the three previous families (fam000574, fam000680, and fam000706) are involved in functions related to cell envelope, we hypothesize they are involved in similar functions.

**Distinction between the CPR and other bacterial symbionts**. Dependentiae are placed within the CPR bacteria in the first dataset (Fig. 3b). In the second dataset, bacteria with highly reduced genomes and Tenericutes are basal to the CPR (Fig. 4b). The clustering of CPR with Dependentiae and bacteria with highly reduced genomes likely occurs because they share the low incidence of many protein families (Supplementary Fig. 15). However, CPR-enriched families are rare in the Dependentiae (average ~10%, but some genomes have none), bacteria with highly reduced genomes, and the Tenericutes (average ~2%; Supplementary Fig. 15).

## Discussion

Genome-resolved metagenomics studies have greatly expanded our understanding of microbial life, particularly through discovery of new bacterial lineages. Lacking have been studies that investigate these genomes from the perspective of the diversity and distribution patterns of homologous proteins. To begin comparing protein sequence inventories, we clustered the amino acid sequences into families that approximate homologous groups. These families serve as a common language that enables comparison of gene inventories within and among lineages. Strikingly, the combinations of protein families associated with widespread biological functions separate the CPR from all other bacteria. In other words, the pattern of presence/absence of relatively widely distributed protein families highlights a major dichotomy within Domain Bacteria that corresponds almost exactly to the subdivide inferred based on phylogenetic analyses (both rRNA and concatenations of ribosomal proteins)[14].

CPR bacteria can be differentiated from other bacteria, including other bacterial symbionts (Supplementary Fig. 15), based on 106 genes that are absent or less abundant in other bacteria. The near ubiquity of these families across the CPR radiation is most readily explained by early acquisition at the time of the origin of CPR, with persistence via vertical inheritance. Based on the functional predictions, the protein families that are enriched in CPR and absent or less abundant in other bacteria may be important for interaction between CPR and their hosts. Among them, the type IV pili may be central to CPR associations with other organisms (Fig. 6). These molecular machines confer a broad range of functions from locomotion, adherence to host cells, DNA uptake, protein secretion and environmental sensing[49]. Notably, several other groups of CPR-enriched genes are also predicted to function in DNA uptake and maintenance of pilin structure. Given the overall small genome size, these findings reinforce the conclusion that genes for organism–organism interaction are central to the lifestyles of CPR bacteria. Future work would help to refine the set of CPR enriched genes thanks to the addition of new genomes and the use of statistical tests that take account of phylogeny.

Within the CPR we identified many clusters of genomes that share similar core metabolic platforms (Supplementary Fig. 12B). Some CPR bacterial phyla have extensive biosynthetic capacities, whereas others have minimal sets of core protein families (Supplementary Fig. 12B)[12]. This may indicate extensive gene loss in some groups. Given the overall phylum-level consistency of the protein family sets, we suspect that major genome reduction events were ancient.

Looking across the entire analysis, the broad consistency in combinations of core protein families within lineages strongly suggests that the distribution of these families is primarily the result of vertical inheritance. Specifically, the patterns of protein family distribution reproduce the subdivision of Bacteria from Archaea and essentially recapitulate many phylum and subphylum groupings.

Collapsing the branches in the cladogram formed from the hierarchical clustering of protein families revealed enormous branch length in the CPR. We interpret this to indicate huge variation in the sets of core protein families across the CPR (Fig. 4c and Supplementary Fig. 9). This could be due to genetic drift that resulted in pseudogene formation and gene loss that erased the phylogenetic signal between distant families. Alternatively, the large scale of the CPR may be the consequence of its long evolutionary history. Arguing for the first case, CPR have small genomes and probable symbiotic lifestyles. Thus, they may be analogous to obligate endosymbionts of Eukaryotes, whose reduced genomes are due to genetic drift and small effective population sizes. Counter to this, CPR bacteria are not known to be common endosymbionts and small population sizes for CPR are unlikely, as they are abundant members of microbial communities from diverse environments[12]. In the second case, diversity in the core protein family platform may have arisen because symbiotic associations with different groups of bacteria selected for larger or smaller requirements for core biosynthetic capacities in the symbiont.

We envision two different scenarios that might explain our observations (Fig. 7). In the first case (Fig. 7a), we imagine that both CPR and non-CPR bacteria emerged from a protogenote community that preceded cellular lineages[50]. In this case, CPR may have arisen and diversified in parallel with non-CPR bacterial groups. Evolutionary innovations that generated new lineages of potential bacterial hosts could have stimulated major divergences within the CPR, essentially the rise of new CPR phyla. Our analyses suggest that this was coupled to selection of a set of core protein families that persisted within phyla. Evidence that would be consistent with this would be the finding that, in most cases, bacteria of a specific CPR phylum associates with the bacteria of the same non-CPR phylum. At this time, there is only one case in which the host of a CPR bacterium is well established[6], so testing this hypothesis will require new data. In the second scenario (Fig. 7b), CPR arose from within a bacterial lineage and this ancestor underwent rapid but heterogeneous patterns of gene loss (examples of two timings are shown). In this case, a huge diversity of potential hosts would exist as CPR diverged, making symbiont-host associations less likely to be phylum specific (as phylum specificity would require that symbioses are linked to a phylum-wide trait). In the extreme case of very recent appearance of CPR from within a relatively modern lineage, followed by extensive gene loss, the CPR protein core protein family sets should be clearly a subset of those from one phylum, class or order. This does not appear to be the case, as we did not observe a set of gene families in CPR that is shared with a single non-CPR phylum. A third possibility in which different CPR phyla arose from different non-CPR bacterial phyla via rapid genome reduction was not considered because of the clear monophyly of the radiation[5,14,15].

The relative magnitude of diversity of distinct core gene sets in the CPR compared to non-CPR bacteria is consistent in scale with the relative magnitude of phylogenetic diversity of these groups, as rendered in ribosomal RNA and protein trees[14]. It is interesting to note that symbiotic associations involving bacterial hosts could have evolved as far back in time as the emergence of bacterial cells, potentially around 3.7 billion years ago[51]. This contrasts with the well-studied symbioses involving eukaryotic hosts, such as insects, which evolved only around 0.5 billion years ago[52]. Thus, we should consider the possibility of a very long evolutionary history for CPR bacteria in symbiotic associations with bacterial hosts.

## Methods

**Dataset construction**. The initial dataset contains 3,598 prokaryotic genomes (5,061,957 proteins) that were retrieved from four published datasets[5,9,17,18]. The dataset encompasses 2321 CPR (1,953,651 proteins); 1198 non-CPR bacteria (3,018,597 proteins) and 79 Archaea (89,709 proteins) (Supplementary Data 2). The second 'NCBI' dataset contains 2729 genomes (8,425,478 proteins). Genomes were chosen based on the taxonomy provided by the NCBI. Briefly, for each prokaryotic phylum, one genome per genus was randomly selected from the NCBI genome database (last accessed in December 2017). Some genomes do not have genus assignment although they have a phylum assignment. In those cases, five genomes per phylum were randomly selected. Refseq were preferred to non-refseq genomes as these are generally better annotated. The NCBI dataset encompasses 282 CPR (217,728 proteins); 2278 non-CPR bacteria (7,811,207 proteins) and 169 Archaea (396,543 proteins) (Supplementary Data 2).

**Protein clustering**. Protein clustering into families was achieved using a two-step procedure (Supplementary Fig. 1). A first protein clustering was done using the fast and sensitive protein sequence searching software MMseqs2 (version 9f493f538d28b1412a2d124614e9d6ee27a55f45)[53]. An all-vs.-all search was performed using e-value: 0.001, sensitivity: 7.5, and cover: 0.5. A sequence similarity network was built based on the pairwise similarities and the greedy set cover algorithm from MMseqs2 was performed to define protein subclusters. The resulting subclusters were defined as subfamilies. In order to test for distant homology, we grouped subfamilies into protein families using an HMM–HMM comparison procedure as follows. The proteins of each subfamily with at least two protein members were aligned using the result2msa parameter of mmseqs2, and

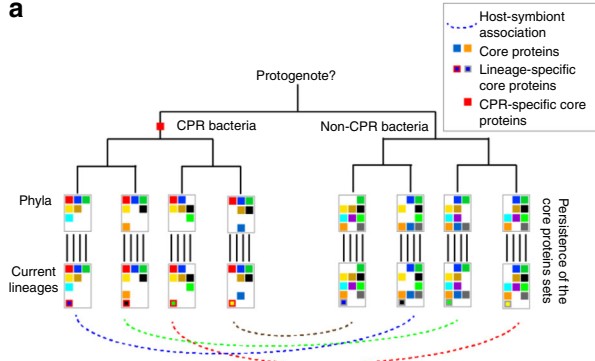

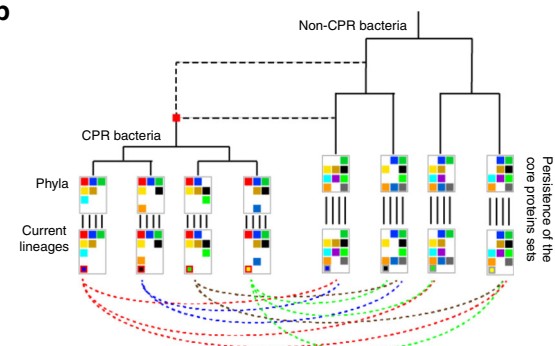

**Fig. 7** Two scenarios for the origin and the evolution of the CPR. **a** In the first scenario, CPR and non-CPR bacteria emerged from the protogenote community and co-evolved. In this case, major divergences within the CPR, essentially the rise of new CPR phyla, may have been stimulated by evolutionary innovations that generated new lineages of potential bacterial hosts. **b** In the second scenario, CPR evolved from within the non-CPR bacteria and experienced a huge genome reduction

from the multiple sequence alignments HMM profiles were built using the HHpred suite (version 3.0.3)[54]. The subfamilies were then compared to each other using hhblits[55] from the HHpred suite (with parameters -v 0 -p 50 -z 4 -Z 32000 -B 0 -b 0). For subfamilies with probability scores of ≥95% and coverage ≥0.50, a similarity score (probability × coverage) was used as weights of the input network in the final clustering using the Markov CLustering algorithm[56], with 2.0 as the inflation parameter. These clusters were defined as the protein families.

**Selection of widespread families**. Examining the distribution of the protein families across the genomes, a clear modular organization emerged (Fig. 2a). We used the Louvain algorithm[23] to detect modules of proteins that share similar patterns of presence/absence across the genomes. Briefly, the Louvain algorithm seeks a partition of a network that maximizes the modularity index Q. The algorithm was performed on a weighted network that was built by connecting family nodes sharing a Jaccard index >0.4. For each pair of protein families, the Jaccard index was calculated based on their profiles of presence/absence across the genomes. The 0.4 threshold was empirically chosen because it defined three distinct modules for widespread proteins in Archaea, non-CPR bacteria and bacteria (see Fig. 2a) whereas lower thresholds merged families having distinct presence/absence patterns across the genomes. This procedure defined modules with more than 10 proteins.

A Phyla distribution was assigned to each module. Because modules contain genomes that carry only few families of the modules, we designed a procedure to only identify genomes that carry most of the families of the modules. For each module, the median number of genomes per family ($m$) was calculated. The genomes were ranked by the number of families they carry. The $m$ genomes that carry the most of families were retained; their phyla distribution defines the taxonomic assignment of the module.

**Hierarchical clustering of the genomes and the families**. The genomes were hierarchically clustered using the Jaccard distance that was calculated based on profiles of protein family presence/absence. The families were also hierarchically clustered based on profiles of presence/absence in genomes. We used an agglomerative (also called bottom-up) method for the hierarchical clustering. In agglomerative clustering, we assign each observation to its own cluster (step 1).

Then, we compute the similarity (e.g., distance) between each of the clusters (step 2) and then join the two most similar clusters (step 3). Steps 2 and 3 are repeated until there is only a single cluster left. Agglomerative clustering can use various measures to calculate the distance between two clusters. Three different measures were used: single linkage, complete linkage, and average linkage. In single-linkage hierarchical clustering, the distance between two clusters is defined as the shortest distance between two points in each cluster. In complete-linkage hierarchical clustering, the distance between two clusters is defined as the longest distance between two points in each cluster. In average-linkage hierarchical clustering, the distance between two clusters is defined as the average distance between each point in one cluster to every point in the other cluster.

**Genome completeness assessment and de-replication**. Genome completeness and contamination were estimated based on the presence of single-copy genes (SCGs) as described in ref. [9]. For CPR, 43 universal SCGs were used, following[9]. In non-CPR bacteria, genome completeness was estimated using 51 SCGs, following[9]. For archaea, 38 SCGs were used, following ref. [9]. Genomes with completeness >70% and contamination <10% (based on duplicated copies of the SCGs) were considered as draft-quality genomes. Genomes were de-replicated using dRep[57] (version v2.0.5 with ANI >95%). The most complete genome per cluster was used in downstream analyses.

**Functional annotation**. Protein sequences were functionally annotated based on the accession of their best Hmmsearch match (version 3.1) (E-value cut-off 0.001)[58] against an HMM database constructed based on ortholog groups defined by the KEGG[20] (downloaded on June 10, 2015). Domains were predicted using the same Hmmsearch procedure against the Pfam database (version 31.0)[59]. The domain architecture of each protein sequence was predicted using the DAMA software (version 1.0) (default parameters)[60]. SIGNALP (version 4.1) (parameters: -f short -t gram+)[61] and PSORT (version 3.0) (parameters:–long–positive)[61] were used to predict the putative cellular localization of the proteins. Prediction of transmembrane helices in proteins was performed using TMHMM (version 2.0) (default parameters)[62]. The transporters were predicted using BLASTP (version 2.6.0)[63] against the TCDB database (downloaded on February 2019) (keeping the best hit, e-value cut-off 1e-20)[36].

**Phylogenetic tree reconstruction**. The two maximum-likelihood trees were calculated based on the concatenation of 14 ribosomal proteins (L2, L3, L4, L5, L6, L14, L15, L18, L22, L24, S3, S8, S17, and S19) using RAxML (version 8.2.10)[64] (as implemented on the CIPRES web server[65]), under the LG plus gamma model of evolution (PROTGAMMALG in the RAxML model section), and with the number of bootstraps automatically determined.

**Detection of the families in the second 'NCBI' dataset**. For the second 'NCBI' dataset, a database of all HMMs of the subfamilies was created to identify members of each family in the second 'NCBI' dataset. Protein sequences were annotated based on the subfamily of their best HMM score using Hmmsearch (version 3.1) (E-value cut-off 0.001, coverage threshold of the HMM >0.5) against the HMM database of subfamilies.

**Enrichment analysis**. Enrichment/depletion of protein families was calculated based on the frequency of the computed protein families in the first and second 'NCBI' datasets. The enrichment of each family in CPR vs. non-CPR bacteria was computed using a Fisher's exact test on a contingency table of presence/absence in CPR and non-CPR bacteria genomes. Families were considered enriched or depleted if their p-values, after correction for false detection rate (Benjamini-Hochberg), were significant ($<10^{-5}$) in both datasets. The remaining families were assigned as equally distributed.

**Reporting Summary**. Further information on research design is available in the Nature Research Reporting Summary linked to this article.

## Data availability

All genomes used in the analysis are publicly available (see Supplementary Data 2). The fasta sequences of the 921 families and the binary matrices used to create Figs. 2–4 are available from figshare at https://doi.org/10.6084/m9.figshare.6296987.

## Code availability

Scripts used to perform the protein clustering are available at https://github.com/raphael-upmc/proteinClusteringPipeline.

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

## Acknowledgements

Support was provided by grants from the Lawrence Berkeley National Laboratory's Genomes-to-Watershed Scientific Focus Area. The U.S. Department of Energy (DOE), Office of Science, and Office of Biological and Environmental Research funded the work under contract DE-AC02-05CH11231 and the DOE carbon cycling program DOE-SC10010566, the Innovative Genomics Institute at Berkeley and the Chan Zuckerberg Biohub. D.B. was supported by a long-term EMBO fellowship.

## Author contributions

R.M., D.B., C.C, and JFB designed the analysis. RM designed the final clustering method, assembled the second dataset, performed clustering analyses, detected the widespread families, created the binary matrix and performed the functional analysis. DB assembled the initial dataset and performed the initial protein clustering. CC performed the phylogenetic tree and contributed to functional analyses. All authors contributed to the analysis of the data and the interpretation of the results. RM and JFB wrote the manuscript. All authors read and approved the final manuscript.

## Additional information

**Competing interests:** JFB is a founder of Metagenomi. The other authors declare no competing interests.

