## [Peer Review File · Nature Communications]

Reviewers' Comments:

Reviewer #1:

Remarks to the Author:

The manuscript by Méheust and colleagues compares presence and absence patterns of homologous genes across bacteria and archaea. The aim was to better characterize the candidate phyla radiation in terms of the function and evolution of its members. The candidate phyla radiation is a current and exciting topic.

Many genomes are available, all of uncultured organisms. So far, the function of CPR bacteria was mainly investigated by looking at genes known to be important for well studied bacteria.

Unfortunately, such genes were often absent. The present approach is agnostic and unbiased by preconceptions about the importance of genes. Therefore, the approach makes much sense to me.

Based on the data, the authors infer that CPR bacteria are distinct from other bacteria and archaea in terms of their gene content. They were also shown to be quite different from each other based on gene content, indicating a lot of variety.

The authors found that part of the genes characteristic of CPR were associated with functions such as transport and attachment. That is consistent with the symbiotic lifestyle that has so far been assumed to define the radiation. Finally, the authors speculate about CPR evolution. Here the novel finding and conclusion is that many CPR genes appear to be inherited horizontally, suggesting a monophyletic origin.

I think this manuscript is well written, the storyline was easy to follow. The materials and methods section is clear and the authors provide a link to the data behind the figures as well as a fasta file with the clustered genes, enabling others to reproduce and further explore the data.

My only criticism is that I would like to see the authors provide the rationale for their approach (toward the end of the introduction) better-rooted in the discipline of phylogenomics, e.g. the work of Dutilh and others. There is quite a lot of debate on use of homologues versus orthologues, etc, which I think the authors should not ignore; they should motivate their methodological choices in this context. Also (very minor point), I think the mcl manual suggests use of e-values, not % ids for clustering. I'm curious whether the authors compared these approaches? If they would have clustered by e-value, N-terminal and C-terminal extensions might not have produced the noisiness they describe.

It would be nice if the authors could expand a bit on the putative transporter genes they found? What type of transporters, and what might they transport? Even if this remains unknown, in view of the presumed symbiotic lifestyle, transporters are interesting.

Did patterns of gene occurrence provide any new hints on differences in lifestyle between CPR phyla?

I think the large differences in gene content might also be explained by a lack of genome streamlining, as it remains unknown to what extent the genes are actually important to their hosts. Or did the authors falsify that possibility?

There is a minor typo on p7 line 155 ("to" basic metabolism) that leads to a grammar error.

Marc Strous

Reviewer #2:
Remarks to the Author:
Major comments

This paper describes the comparison of the biological capacities found across bacteria (and a few archaea) based on the clustering by genome content of the different species represented in a very large reconstructed dataset of CPR and non-CPR genomes. The authors argue that bacteria members of the newly discovered, very interesting, and largely elusive "Candidate Phyla Radiation" (CPR), have very different genome content, and therefore, distinct biological capacities when compared to other non-CPR bacteria (and archaea).

The manuscript is well-written, and very clear. While the results of the genome-content clustering are interesting, and seem to support rather unique capacities for CPR bacteria, it seems to me that the biological insights are limited, and that the clustering approach could have been more thorough, as the entire manuscript relies on it:

1) the particular potential capacities of CPR are discussed superficially. Most discussion focuses on type IV pili and their particular pilins (section starting l. 294, l. 385). But T4P are known to be widespread in bacteria, and pilins are known to evolve rather quickly and thus harbor high sequence diversity, which makes the "novel" pilins not that surprising (Pelicic, *Molecular Microbiology* 2008, Giltner et al., *MMBR* 2012). In the end, "only" 87 families are proposed to be enriched in CPR over non-CPR (l. 377-379). It is not a lot when compared to the claim of large biological capacities differences between the two groups. Also, could't the "lack" of biological capacities unites the CPR against the others (lines 374-376) (see below)? The 453 "CPR-depleted" families are too little discussed to my taste. They might also point at commonalities between CPR.

2) the positioning of a non-CPR lineage (TM6, line. 174) within CPR based on genome-content genomes clustering is not really discussed, and the gene repertoire observed for this lineage, which seems rather "non-CPR"-like (e.g. on Fig 3A) raises concern on the clustering approach used (see below).

3) the fact that genome-content-based phylogenies recapitulate molecular sequences phylogenies has been known for many years (Snel 1999, *Nature Genetics*), and it is not that surprising that CPR are separated from non-CPR both based on both phylogenetics and genome-content-based approaches, even though this is repeatedly claimed (lines 181, 186-187, 217, 404-408). What is surprising in the authors' results is the extent of the difference in the patterns of genome content between CPR and non-CPR.

Follow more detailed comments:

- Results on Fig. 2A do not seem to show many specificities in terms of genome content for CPR versus non-CPR, aside from the fact that they have much less families (but maybe this is only a visual effect, families and genomes could maybe be arranged differently on the figure).

Also, I think that by definition, the choice of the "modules" increases the apparent differences in genome-content between CPR and non-CPR (lines 141-144). Thus, my question is: what would look the Fig 4C like if it were based on all families?

- Regarding the genome clustering, it would have been interesting to see how looks the clustering of non-CPR bacteria alone, and CPR bacteria alone. Given the reduced nature of the CPR genomes (and maybe because of sequence divergence as well), they have lots of families absent, and therefore, could generate very high distances with other genomes (outliers, long branches). I wonder how this

influences the hierarchical clustering, known to be susceptible to long branch attractions especially as it is based here on complete linkage (or "maximum distance" linkage). Given the nature of the data, the average or centroid linkage could be less susceptible to outliers than the complete linkage. Have the authors tried several clustering approaches? This is crucial, as the paper results heavily rely on it.

- A comment related to the above: TM6, a non-CPR, is an outlier and groups with CPR... Also, based on Fig. 3A (even though it is too tiny to be sure...), TM6 genome-content data seem to form a line that diverges from that of the other CPRs, making this positioning suspicious. Would TM6 positioning be different with another clustering method? The authors did not discuss that, or the potential biological reasons why TM6 could be more similar to CPR than to non-CPR. The fact that TM6 do not have the CPR-enriched families (line 175) is actually not really reassuring to me, as for example extensive transfers of CPR-specific families could have been a potential reason for their grouping with CPR.

- For genome-content clustering, a network-based clustering, as the one used for the families clustering could have been complementary, and interesting to look into, especially e.g., when it comes to TM6's position.

- lines 390-398: I find odd this line of discussion. It is well-known that sets of lineage-specific core genes are conserved along evolution via vertical transmission, and that species sharing ancestors to the exception of other species harbor more shared traits/genes than other. Also, the discussion of Cyanobacteria having clear conserved traits with respect to their particular physiology, when other bacterial phyla not, seems rather to be some kind of bias in biological knowledge. I would remove, or at least re-shape this section.

- lines 408-410: "Collapsing... huge variation in the sets of core protein families" I think the pattern observed in fig. 4C could also be the result of fast sequence evolution, and therefore less shared families. Could the authors discuss this?

- Fig. S6A and S6C: What does represent the Y axis exactly? Proportion of families? Should'nt they add up to 1 by functional category then, between depleted, enriched, and equally distributed?

- I did not fully understand the principles of the enrichment analysis (Methods)... Why involve ribosomal proteins?

Minor comments

- line 99: cite the paper introducing the 16 marker genes.

- line 101: "of" missing

- line 143: say already here how many families accounted for by the 6 selected modules.

- line 171: explain already here the clustering method used: "hierarchical clustering, distance ..., ..." (only mentioned in Fig. 3 legend, and further down). Plus maybe also in Methods?

- line 346: 76, out of how many CPR genomes?

- Fig. 2A and 3A: put title on axes.

- Fig. 3A: same color (red) for different purposes: function category and "CPR". Choose distinct colors?

- Fig S6B is not super informative without any details (color-code?) on the functional categories.

- Revise references: encodage and numbering.

I know supplementary materials are not reviewed, but here are some small suggestions/comments:

- Fig. S1 legend: add the function of the families.

- Fig. S6: panels letter inverted (B and C).

Reviewer #3:

Remarks to the Author:

This manuscript describes a very large scale analysis of protein clustering into families in Bacteria. It contrasts the results for CPR bacteria (a large clade of uncultivable bacteria discovered a few years ago by this lab), and the other Bacteria (there are also some Archaea in the analysis, but they are not the focus of the study). Most of the analyses concern a small subset where the authors pick half a dozen groups that (with one exception) have many more elements on one group than on the other. The two key claims of the study are that gene repertoires follow the phylogeny of Bacteria, and that CPR and non-CPR use very different sets of genes. The paper is well-written and the topic is very interesting. But the first claim is perfectly in line with the large literature on molecular evolution and the analysis is prone to biases that may spuriously favor the second claim.

Gene families follow phylogeny. The authors find these results remarkable. I guess that may depend on the perspective, but to me they seem trivial because this is the null hypotheses in most studies (and has been verified many times). This will naturally occur for three reasons (all with well-demonstrated examples within non-CPR). First, phylogeny matters because genes are transferred vertically most of the times. This implicates that bacteria in the same clades have more similar sets of genes. Second, closely related bacteria tend to have similar traits and are more likely to live in similar environments. They will tend to select for the same types of genes. Third, most proteins evolve fast enough that one cannot identify homology between distant clades using blast—like approaches. This means that a clustering method (and this one poses specific problems, see below) will tend to split many protein families in several groups, which will follow phylogenetic lines. This produces a bias: more distant clades will look like having more different gene families, even if they don't. Since CPR form a more or less monophyletic clade at the basis of the bacterial tree, the observation that gene families follows phylogeny cannot be rigorously assessed from this study because the methods are not appropriate for this specific purpose. That would have required more sensitive, precise, and phylogenetically informed methods for the definition of orthologous families.

The second claim is that CPR and non-CPR have very different types of genes. This may be true, but the bias in the analysis (mentioned above) will lead automatically to this conclusion because of CPR monophyly at the basis of the tree and its probable fast evolution (since their genomes are reduced). The method used to make the analysis of the functions that sustain the claim is also not satisfactory. There are many clusters of families in the dataset (156). But at the end, only 6 are analyzed and all but one are all either very abundant in one group or in the other. Naturally, this increases the impression that many genes are present in either one group or the other. For this claim to be convincing (and if it is convincing, then interesting) one needs: a method that is better at identifying distant relations of homology, a control that proteins in the different families are not homologous (the HMM-HMM alignments could be helpful here), and a fairer analysis of the dataset that uses all the data, not just 6 of the 156 clusters. Finally, I think that an analysis based directly on protein families (and not clusters of families) would be more appropriate because that is the appropriate evolutionary and functional unit. In this particular case, the use of Louvain clustering obscures results because clustering is affected by phylogeny.

The method used to protein families is not very well-suited to this analysis. This is because it uses methods which are not very sensitive at this scale (ublast, less sensitive than blastp, itself much less than HMM-based approaches), no information on phylogeny for clustering, and clustering using MCL that is known to produce many very small clusters. This will increase the bias mentioned above: very distant orthologous proteins will be split apart and this will follow phylogeny. To prove the points that

the authors want to make, this initial analyses requires more controls for the lack of sensitivity of ublast and test that clustering by MCL is not producing small clusters. (To be fair, there is a quality clustering analysis using ribosomal proteins, but these are amongst the slowest evolving proteins on earth. A control with fast evolving proteins is more pertinent to sustain the claim about specific functions in each group.)

In the beginning of the study there are >20 000 protein families. But most of the analysis actually concerns the 540 families very abundant in CPR or nonCPR. Of these, 453 are more abundant in nonCPR Bacteria. This is a very important piece of information that shows that CPR do not have that many specific protein families. The <90 families that might be more abundant in nonCPR should be queried to check they have no homologs in the other bacteria with a more sensitive method. This may lead to a number much lower than 90.

If CPR systematically share the trait of being symbionts, then that will increase the likelihood that they group together and that they evolve fast. The argument given against the objection that clustering reflects the common trait of symbiosis, which is that known endosymbionts do not group with them, is not very convincing. The latter, like Buchnera, evolve very fast in sequence and will not cluster with the CPR because the sequence analysis method is not sensitive enough and because these bacteria are often a subset of neighboring bacteria (they have a subset of their gene families). The Louvain and the hierarchical clustering methods would not in principle cluster them with bacteria that are so divergent.

Other points:

The analysis depends on a number of choices (programs, parameters). These should be tested more thoroughly because they may affect the conclusions. How important is the default choice of the inflation parameter in MCL ? What is the effect of gene annotation in producing genes of heterogeneous sizes and therefore spuriously increase the number of families in the most poorly sequenced genomes (CPR)? What is the effect of using more sensitive homology detection methods?

I was intrigued by the claim that type IV pili (tIVp) are very abundant in CPR and rare in nonCPR. There's a lot of data showing that these pili are present in most nonCPR Bacteria and in most Archaea. The claim that they are even more abundant in CPR may result from a clustering problem in fast evolving proteins (the problems mentioned above). Pilins evolve fast in sequence and are short, those of Archaea and Bacteria will not group together by blast (but are homologous as revealed by more sensitive methods). My guess is that the CPR and nonCPR pilins do not cluster together by blastp but would cluster together using more sensitive methods. This is the best explanation for several incoherences in the results: the lack of pilin families in nonCPR in Fig 6 (it is well known that most bacteria encode pilins), the presence of prepilin peptidase in both groups (they are more conserved) which would be unexpected if the nonCPR had fewer pili, and the lack in the list of all the tIVp components that evolve slowly.

The section 'functions over-abundant in CPR' is confusing. There are comparisons with functions that seem common in non-CPR Bacteria, leading me to think that this is also a clustering bias and leaving the reader with little information on the extent of the difference between the two groups. To make this clearer one would need to have a precise quantification of the over-abundance of these functions in CPR and nonCPR. I dived into the excel sheet, but that's hard to follow because there are only counts (not controlled for effective sizes nor phylogeny structure).

The analysis of the correspondence between the clustering analysis and the phylogenetic tree (fig 3C) is very approximative (it's just a succinct verbal argument using the colors matching large fractions of

the trees). There are quantitative ways to compare trees that should be used to produce a quantitative (testable) argument.

L313. There seems to be a misunderstanding on what the literature states on the frequency of competent species. Dubnau and others have indicated that few bacteria are known to be naturally transformable. There is no single CPR known to be transformable if one uses the same criterion (experimental verification). But if one uses the criteria of the paper (a pilus plus some competence genes), then many nonCPR Bacteria would be classed as naturally transformable. So, one should not use Dubnau's paper as a confirmation that these genes are rare in nonCPR against the argument based on the presence of the genes in CPR to state that they are abundant in CPR. The argument should be based on a fair comparison on the frequency of these genes in CPR and nonCPR (and accounting for the fact that these genes may produce distinct protein families In the two groups in the clustering procedure).

I didn't understand the meaning nor the precise details of checking 'annotation admixture'.

Data is clustered with hierarchical clustering, MCL and Louvain clustering. But it's unclear why the method, or the focus, changes along the text.

I found Figures 2A, 3A, 4A, 6 and many in additional online material impossible to read out. These are huge displays of black and white points or lines that are not labelled or highlighted enough to be understood.

L234. It's a pity that the HMM-HMM analysis is not done on all the families, but only on the predefined set of 6 clusters. Families in small clusters may match other families (probably the case of pilins). This means that the clustering is not necessarily robust regarding clustering parameters (contrary to what it's stated in the text). More general controls are needed to make a convincing case.

L482. I found this part of the methods confusing. There is one first clustering by MCL. And then a final clustering by MCL. It is not clear exactly what's the input of this final clustering and how the two differ.

L482. I would suggest making HMM-HMM alignments between families (and not just sub-families) to reveal distant homologies.

L498. This explanation lacks rigor and quantification.

L477. Which percent identity threshold?

The list of references has many problems, with volumes, page numbers and sometimes journal names missing. See references 5, 8, 14, 15, 18, 20, 26, 29, 46, 56, etc

REVIEWS NATURE COMMUNICATION

Reviewer #1 (Remarks to the Author):

The manuscript by Méheust and colleagues compares presence and absence patterns of homologous genes across bacteria and archaea. The aim was to better characterize the candidate phyla radiation in terms of the function and evolution of its members. The candidate phyla radiation is a current and exciting topic.

Many genomes are available, all of uncultured organisms. So far, the function of CPR bacteria was mainly investigated by looking at genes known to be important for well studied bacteria. Unfortunately, such genes were often absent. The present approach is agnostic and unbiased by preconceptions about the importance of genes. Therefore, the approach makes much sense to me.

Based on the data, the authors infer that CPR bacteria are distinct from other bacteria and archaea in terms of their gene content. They were also shown to be quite different from each other based on gene content, indicating a lot of variety.

The authors found that part of the genes characteristic of CPR were associated with functions such as transport and attachment. That is consistent with the symbiotic lifestyle that has so far been assumed to define the radiation. Finally, the authors speculate about CPR evolution. Here the novel finding and conclusion is that many CPR genes appear to be inherited horizontally, suggesting a monophyletic origin.

I think this manuscript is well written, the storyline was easy to follow. The materials and methods section is clear and the authors provide a link to the data behind the figures as well as a fasta file with the clustered genes, enabling others to reproduce and further explore the data.

We thank the reviewer (M. Strouss) for his comments.

My only criticism is that I would like to see the authors provide the rationale for their approach (toward the end of the introduction) better-rooted in the discipline of phylogenomics, e.g. the work of Dutilh and others. There is quite a lot of debate on use of homologues versus orthologues, etc, which I think the authors should not ignore; they should motivate their methodological choices in this context.

It is an interesting point, as orthologous genes are more likely to retain ancestral gene function. However, to delineate orthologs from paralogs is challenging in part due to the large degree of evolutionary divergence when analyzing the entire bacterial domain (e.g., due to gene fusion/fission, gene loss, lineage-specific duplication and horizontal gene transfer). To avoid incorrect delineations, we focus on homologous genes. We edited the introduction to explain the rationale for our approach, it now reads: **“Given the large extent of divergence over the**

history of the bacterial domain and difficulties with accurately distinguishing orthologs from homologs, our approach considered homologous protein families”.

Also (very minor point), I think the mcl manual suggests use of e-values, not % ids for clustering. I'm curious whether the authors compared these approaches? If they would have clustered by e-value, N-terminal and C-terminal extensions might not have produced the noisiness they describe.

Reviewer 3 was also concerned about the sensitivity of our previous protein clustering. In response to both reviewers, we substantially modified our protein clustering to be more sensitive (mmseqs2 was used in place of Usearch, see materials and methods). In addition, we now use the probabilities instead of protein identity (PID) for the HMM-HMM clustering. We choose the probability score over other metrics because according to the hhblits manual: *“The estimated probability of the template to be (at least partly) homologous to your query sequence is the most important criterion to decide whether a template HMM is actually homologous or just a high-scoring chance hit. When it is larger than 95%, say, the homology is nearly certain.”*. This change resulted in the inclusion of almost all the N-terminal and C-terminal extensions into the main ribosomal protein families and so less fragmented families (Table S1).

It would be nice if the authors could expand a bit on the putative transporter genes they found? What type of transporters, and what might they transport? Even if this remains unknown, in view of the presumed symbiotic lifestyle, transporters are interesting.

We used the transporter database TCDB (www.tcdb.org) to annotate our dataset and found some interesting results related to the absence of certain transporters in CPR. We added a new paragraph in the “Biological capacities explain the singularity of CPR” on this topic.

Did patterns of gene occurrence provide any new hints on differences in lifestyle between CPR phyla?

We include a figure and some text that reports that Peregrinibacteria, for example, have more extensive capacities than other CPR.

I think the large differences in gene content might also be explained by a lack of genome streamlining, as it remains unknown to what extent the genes are actually important to their hosts. Or did the authors falsify that possibility?

Indeed, our results cannot rule out this possibility, we added a brief comment to the discussion.

There is a minor typo on p7 line 155 (“to” basic metabolism) that leads to a grammar error.

Fixed, thank you.

Reviewer #2 (Remarks to the Author):

Major comments

This paper describes the comparison of the biological capacities found across bacteria (and a few archaea) based on the clustering by genome content of the different species represented in a very large reconstructed dataset of CPR and non-CPR genomes. The authors argue that bacteria members of the newly discovered, very interesting, and largely elusive "Candidate Phyla Radiation" (CPR), have very different genome content, and therefore, distinct biological capacities when compared to other non-CPR bacteria (and archaea).

The manuscript is well-written, and very clear. While the results of the genome-content clustering are interesting, and seem to support rather unique capacities for CPR bacteria

We thank the reviewer for her/his positive comments.

it seems to me that the biological insights are limited, and that the clustering approach could have been more thorough, as the entire manuscript relies on it:

We re-did the entire protein clustering method and provide extensive description of the approach.

1) the particular potential capacities of CPR are discussed superficially. Most discussion focuses on type IV pili and their particular pilins (section starting l. 294, l. 385). But T4P are known to be widespread in bacteria, and pilins are known to evolve rather quickly and thus harbor high sequence diversity, which makes the "novel" pilins not that surprising (Pelicic, Molecular Microbiology 2008, Giltner et al., MMBR 2012). In the end, "only" 87 families are proposed to be enriched in CPR over non-CPR (l. 377-379). It is not a lot when compared to the claim of large biological capacities differences between the two groups. Also, could'nt the "lack" of biological capacities unites the CPR against the others (lines 374-376) (see below)? The 453 "CPR-depleted" families are too little discussed to my taste. They might also point at commonalities between CPR.

We now report 106 families enriched in CPR. Importantly, the "novel" CPR pilins clustered now with the other pilins although they are still very divergent. The T4P pilins are still statistically enriched in CPR relative to other bacteria, consistent with an important role in interaction/adhesion and uptake of DNA. We expanded the section on CPR-depleted families by discussing the depletion of numerous transporters that are considered as widespread in non-CPR bacteria while almost absent in CPR. Some specific capacities that are relatively scarce in CPR are information system-related but most are involved in biosynthesis.

2) the positioning of a non-CPR lineage (TM6, line. 174) within CPR based on genome-content genomes clustering is not really discussed, and the gene repertoire observed for this lineage,

which seems rather "non-CPR"-like (e.g. on Fig 3A) raises concern on the clustering approach used (see below).

This is partially explained by the fact that many more families are depleted in both CPR and TM6 than in other bacteria, and the distance metric mostly reflects this. However, we consider this to be misleading because when investigated further, it is apparent that the TM6 have far fewer enriched families than occur CPR (Figure S10). We added a paragraph to discuss the placement of TM6 and some reduced-genomes in the light of their genome content, and new supplementary figure.

3) the fact that genome-content-based phylogenies recapitulate molecular sequences phylogenies has been known for many years (Snel 1999, Nature Genetics), and it is not that surprising that CPR are separated from non-CPR both based on both phylogenetics and genome-content-based approaches, even though this is repeatedly claimed (lines 181, 186-187, 217, 404-408). What is surprising in the authors' results is the extent of the difference in the patterns of genome content between CPR and non-CPR.

We concur, and edited the ms to make this statement clearer and now emphasize the notable extent of difference in genome content of CPR and non-CPR bacteria.

Follow more detailed comments:

- Results on Fig. 2A do not seem to show many specificities in terms of genome content for CPR versus non-CPR, aside from the fact that they have much less families (but maybe this is only a visual effect, families and genomes could maybe be arranged differently on the figure).

Figure 2A (distribution of 22,977 families across 2,890 genomes) provides an overview of the distribution of families and is included for completeness.

Also, I think that by definition, the choice of the "modules" increases the apparent differences in genome-content between CPR and non-CPR (lines 141-144). Thus, my question is: what would look the Fig 4C like if it were based on all families?

We think that the choice of the "modules" may have had the opposite effect. The modules (921 families) were chosen because they are comprised of predicted proteins that are widespread in both CPR and non-CPR bacteria. The families that we do not focus on tend to be lineages-specific and their inclusion would have increased (not decreased) the distinction between CPR and no-CPR bacteria.

With regards "what would look the Fig 4C like if it were based on all families?", we performed the same genome clustering based on the 22,977 families as we did for the 921 families in the Figure 3B. The genomes trees are very similar, regardless of the clustering method (Figures S4, S5 and S6, correlation between two cophenetic distance matrices of the two trees: average method = 0.92, single method = 0.84 and complete method = 0.88).

We performed the same analysis with the second “NCBI” dataset (Figure 4C), again the genomes trees are very similar, regardless of the clustering methods used (Figure S7, correlation between two cophenetic distance matrices of the two trees: average method = 0.92, single method = 0.88 and complete method = 0.86).

- Regarding the genome clustering, it would have been interesting to see how looks the clustering of non-CPR bacteria alone, and CPR bacteria alone. Given the reduced nature of the CPR genomes (and maybe because of sequence divergence as well), they have lots of families absent, and therefore, could generate very high distances with other genomes (outliers, long branches). I wonder how this influences the hierarchical clustering, known to be susceptible to long branch attractions especially as it is based here on complete linkage (or "maximum distance" linkage). Given the nature of the data, the average or centroid linkage could be less susceptible to outliers than the complete linkage. Have the authors tried several clustering approaches? This is crucial, as the paper results heavily rely on it.

As noted above, we tested three different agglomerative clustering methods (complete-linkage, average-linkage and single-linkage methods). The trees are shown in Figure S4 and S5 for the first dataset. The results are very comparable.

- A comment related to the above: TM6, a non-CPR, is an outlier and groups with CPR... Also, based on Fig. 3A (even though it is too tiny to be sure...), TM6 genome-content data seem to form a line that diverges from that of the other CPRs, making this positioning suspicious. Would TM6 positioning be different with another clustering method? The authors did not discuss that, or the potential biological reasons why TM6 could be more similar to CPR than to non-CPR. The fact that TM6 do not have the CPR-enriched families (line 175) is actually not really reassuring to me, as for example extensive transfers of CPR-specific families could have been a potential reason for their grouping with CPR.

We carefully looked into the details regarding TM6 and the organisms with reduced genomes, and added a new section in the light of their gene content.

- For genome-content clustering, a network-based clustering, as the one used for the families clustering could have been complementary, and interesting to look into, especially e.g., when it comes to TM6's position.

We thank the reviewer for this suggestion. We performed Louvain clustering on a genome network with different parameters and found that TM6 tends to cluster more with non-CPR bacteria than with CPR. We conclude that TM6 is an interesting and distinctive group.

- lines 390-398: I find odd this line of discussion. It is well-known that sets of lineage-specific core genes are conserved along evolution via vertical transmission, and that species sharing ancestors to the exception of other species harbor more shared traits/genes than other. Also, the discussion of Cyanobacteria having clear conserved traits with respect to their particular

physiology, when other bacterial phyla not, seems rather to be some kind of bias in biological knowledge. I would remove, or at least re-shape this section.

We have removed this section.

- lines 408-410: "Collapsing... huge variation in the sets of core protein families" I think the pattern observed in fig. 4C could also be the result of fast sequence evolution, and therefore less shared families. Could the authors discuss this?

We now discuss this possibility in the ms (also see response to Reviewer 1)

- Fig. S6A and S6C: What does represent the Y axis exactly? Proportion of families? Should'nt they add up to 1 by functional category then, between depleted, enriched, and equally distributed?

The Y-axis represents the proportion of families in the depleted, enriched and equally distributed subsets of widespread families.

We originally wanted to compare the KEGG functional distribution for each of these 3 subsets. For clarity, we have now split each barplot into three subplots (enriched, depleted, equally_distributed) and use the counts instead of the proportions.

- I did not fully understand the principles of the enrichment analysis (Methods)... Why involve ribosomal proteins?

We formerly considered that because the 16 ribosomal proteins are universal genes in both CPR and non-CPR Bacteria, families that have a similar distribution to the ribosomal proteins in CPR and non-CPR will be considered as equally distributed. However, we realized that it is not the best way to detect differences between two groups. Thus, we now rely on a Fisher's exact test on a contingency table of presence/absence in CPR and non-CPR bacterial genomes for each family. Families were considered enriched or depleted if their adjusted p-values, after correction for false detection rate (Benjamini-Hochberg), were significant ($< 10^{-5}$) in both datasets. The remaining families were classified as equally distributed. We provided in the Table S2 the contingency table, p-value, adjusted p-value and odd ratio for the widespread (921) families and both datasets. We edited the method section accordingly.

Minor comments

- line 99: cite the paper introducing the 16 marker genes. - done

- line 101: "of" missing - done

- line 143: say already here how many families accounted for by the 6 selected modules. - done

- line 171: explain already here the clustering method used: "hierarchical clustering, distance ..., ..." (only mentioned in Fig. 3 legend, and further down). Plus maybe also in Methods? - done

- line 346: 76, out of how many CPR genomes? - done

- Fig. 2A and 3A: put title on axes. - done
- Fig. 3A: same color (red) for different purposes: function category and "CPR". Choose distinct colors? - we prefer to retain the current coloring scheme
- Fig S6B is not super informative without any details (color-code?) on the functional categories.
- done
- Revise references: encodage and numbering. - done

I know supplementary materials are not reviewed, but here are some small suggestions/comments:

- Fig. S1 legend: add the function of the families.
- Fig. S6: panels letter inverted (B and C).

We edited the Figure S6 (now Figure S9). Regarding the Figure S1, we guess the referee referred to another figure?

Reviewer #3 (Remarks to the Author):

This manuscript describes a very large scale analysis of protein clustering into families in Bacteria. It contrasts the results for CPR bacteria (a large clade of uncultivable bacteria discovered a few years ago by this lab), and the other Bacteria (there are also some Archaea in the analysis, but they are not the focus of the study). Most of the analyses concern a small subset where the authors pick half a dozen groups that (with one exception) have many more elements on one group than on the other.

We assume the reviewer is referring to the modules comprised of 921 families. These were chosen because they are widespread in CPR and non-CPR bacteria, not because "they have more elements in one group than another".

The two key claims of the study are that gene repertoires follow the phylogeny of Bacteria, and that CPR and non-CPR use very different sets of genes. The paper is well-written and the topic is very interesting.

We thank the reviewer for these positive comments.

But the first claim is perfectly in line with the large literature on molecular evolution and the analysis is prone to biases that may spuriously favor the second claim.

We have substantially modified the manuscript to incorporate the literature on molecular evolution and substantially changed our focus to other topics. Regarding the biases, we address these points below.

Gene families follow phylogeny. The authors find these results remarkable. I guess that may depend on the perspective, but to me they seem trivial because this is the null hypotheses in most studies (and has been verified many times). This will naturally occur for three reasons (all with well-demonstrated examples within non-CPR). First, phylogeny matters because genes are transferred vertically most of the times. This implicates that bacteria in the same clades have more similar sets of genes.

We agree with this point and emphasize that our findings (based on a substantially enlarged dataset due to dramatically expanded sampling of candidate phyla) support the prior results.

Second, closely related bacteria tend to have similar traits and are more likely to live in similar environments. They will tend to select for the same types of genes.

At the phylum level, we suggest that this is not likely. Different members of a single phyla could occupy potentially the full range of Earth's habitat types.

Third, most proteins evolve fast enough that one cannot identify homology between distant clades using blast-like approaches. This means that a clustering method (and this one poses

specific problems, see below) will tend to split many protein families in several groups, which will follow phylogenetic lines. This produces a bias: more distant clades will look like having more different gene families, even if they don't. Since CPR form a more or less monophyletic clade at the basis of the bacterial tree. In, the observation that gene families follows phylogeny cannot be rigorously assessed from this study because the methods are not appropriate for this specific purpose. That would have required more sensitive, precise, and phylogenetically informed methods for the definition of orthologous families.

The major point raised by this referee relates to the lack of sensitivity of the protein clustering and the lack of quality control. It is critical to note that two important points that perhaps we not sufficiently clearly stated in the original manuscript. The reviewer states below that we did not use HMM-HMM comparisons, but this is wrong. In fact it is precisely the method used. The reviewer also notes that we only use of ribosomal proteins to assess the quality of the clustering and did not use a systemic approach using KEGG annotations. In fact, we used both methods.

In fact, we completely revised the methods and find that the conclusions are not substantially different, although stronger. Specifically, we created a more sensitive new protein clustering pipeline especially in replacing usearch by mmseqs2 (as the referee raised a concern regarding the lack of sensitivity of usearch). Mmseqs is as sensitive as blastp, we discussed this further below. We also tuned several thresholds in order to be more inclusive (in reducing the coverage threshold and the probability threshold in the HMM-HMM comparison step). We re-did all the protein clustering analysis and subsequent analysis. The sensitivity and inclusivity of the pipeline have been greatly improved as 4,449,296 out of 5,061,957 protein sequences were clustered into 22,977 families (≥ 5 distincts genomes) while the previous pipeline clustered 3,308,983 protein sequences into 21,612 families (≥ 5 distinct genomes). Of note, we detected more widespread families 921 families are now reported as widespread instead of the 786 in the first version of the manuscript.

The second claim is that CPR and non-CPR have very different types of genes. This may be true, but the bias in the analysis (mentioned above) will lead automatically to this conclusion because of CPR monophyly at the basis of the tree and its probable fast evolution (since their genomes are reduced). The method used to make the analysis of the functions that sustain the claim is also not satisfactory.

Please see response above.

There are many clusters of families in the dataset (156). But at the end, only 6 are analyzed and all but one are all either very abundant in one group or in the other. Naturally, this increases the impression that many genes are present in either one group or the other.

Please see the explanation above.

For this claim to be convincing (and if it is convincing, then interesting) one needs: a method that is better at identifying distant relations of homology, a control that proteins in the different families are not homologous (the HMM-HMM alignments could be helpful here),

See above regarding the use of HMM-HMM alignments.

and a fairer analysis of the dataset that uses all the data, not just 6 of the 156 clusters.

As noted above, we did not include the other 232 modules because of their sparse distributions across the genomes. We now explain that we did not consider the 5 modules only from Archaeal genomes, 53 modules only found in non-CPR genomes and the 171 modules only found in CPR genomes. The 2 remaining modules (11 and 14 families each) are found in a mixture of CPR and non-CPR genomes but they were only found in genomes from 2 and 4 phyla, respectively. This is illustrated in the Figure 2B (blue dots).

Finally, I think that an analysis based directly on protein families (and not clusters of families) would be more appropriate because that is the appropriate evolutionary and functional unit. In this particular case, the use of Louvain clustering obscures results because clustering is affected by phylogeny.

It is important to note that modules were only used to identify protein families that are widespread in both CPR and non-CPR bacteria (and the Louvain clustering was only used in this context). We tried several algorithms such as biclustering and bipartite graph clustering with poor results. Only the Louvain clustering (a popular algorithm with >8581 citations), an unsupervised method, identified modules that were consistent with what those evident in Figure 2A.

Overall, the analysis does focus on protein families, not just clusters of families. We have clarified this in the revised manuscript. This approach lead to consideration of >3 million protein sequences, of a total of ~ 5 million in the analysis. The manuscript has been modified to state this.

The method used to protein families is not very well-suited to this analysis. This is because it uses methods which are not very sensitive at this scale (ublast, less sensitive than blastp, itself much less than HMM-based approaches),

Please see responses to the same misunderstanding provided above

no information on phylogeny for clustering,

We apologize, but this comment is unclear. There is information provided regarding phylogeny throughout the manuscript.

and clustering using MCL that is known to produce many very small clusters. This will increase the bias mentioned above: very distant orthologous proteins will be split apart and this will follow phylogeny. To prove the points that the authors want to make, this initial analyses requires more controls for the lack of sensitivity of ublast and test that clustering by MCL is not producing small clusters. (To be fair, there is a quality clustering analysis using ribosomal proteins, but these are amongst the slowest evolving proteins on earth. A control with fast evolving proteins is more pertinent to sustain the claim about specific functions in each group.)

We agree with the referee that ublast is less sensitive than blastp and have changed our methods, as noted above (although the results are not substantially different). The new pipeline replaces ublast with mmseqs2, a recently published method with similar sensitivity of blastp (when sensitivity threshold is set to 7.5) while being 40 time faster. The speed is critical to allow the clustering of millions of protein sequences in a reasonable time.

We disagree with the referee regarding the fact that “*clustering using MCL that is known to produce many very small clusters*” as the size of the clusters depends of the inflation parameter chosen (the bigger the inflation parameter the greater the number of clusters; see below regarding the MCL inflation parameter).

We agree with the referee that the ribosomal proteins are among the slowest evolving proteins on earth and are not the best representative sets of proteins to assess the quality of the protein clustering. Nevertheless, the referee did not consider the fact that we did performed the same quality control on non-ribosomal proteins by systematically verifying that the protein family groupings approximate KEGG functional annotations. Please see the second paragraph of the result section and FigureS 2B (we apologize that this was wrongly named S1B in the previous version ms). The KEGG annotation in our dataset encompasses 7700 unique annotations from various biological processes including fast evolving defense mechanisms. For each of these 7700 annotations, we reported the family which contains the highest ratio of proteins annotated with that KEGG accession. As expected, the protein clustering is inferior when clustering of the ribosomal proteins but it good for 89.2% of the annotations (6872 out of 7700), as one family always contained >80% of the proteins with a given KEGG annotation. For clarity, we edited the complete paragraph, replaced the figure S2B and edited its legend.

In the beginning of the study there are >20 000 protein families. But most of the analysis actually concerns the 540 families very abundant in CPR or nonCPR. Of these, 453 are more abundant in nonCPR Bacteria. This is a very important piece of information that shows that CPR do not have that many specific protein families. The <90 families that might be more abundant in nonCPR should be queried to check they have no homologs in the other bacteria with a more sensitive method. This may lead to a number much lower than 90.

We performed the HMM-HMM comparison using two different datasets, and have now searched with a more sensitive method. The new clustering pipeline and a new statistical test requested by Reviewer 2 allowed the detection of 106 families enriched in CPR.

If CPR systematically share the trait of being symbionts, then that will increase the likelihood that they group together and that they evolve fast. The argument given against the objection that clustering reflects the common trait of symbiosis, which is that known endosymbionts do not group with them, is not very convincing. The latter, like Buchnera, evolve very fast in sequence and will not cluster with the CPR because the sequence analysis method is not sensitive enough and because these bacteria are often a subset of neighboring bacteria (they have a subset of their gene families).

The fast-evolving nature of the CPR is sometimes stated, but not proven. Although many CPR are likely symbionts, they are not endosymbionts. Endosymbionts like Buchnera are subject to genetic drift due to their small population size and this explains their fast-evolving nature. Small population size in CPR is far from certain, as they can be both abundant and diverse in many environments. Unlike well described symbionts (e.g., of insects) these organisms are not placed on long branches within a radiation of “normal” bacteria. Rather, the entire radiation has similar and features. It is one possibility that they have adopted symbiotic lifestyle (e.g., with other specific bacteria, as is the case for TM7) from the early period of cellular life. We also note that a reduced genome can be the result of streamlining as streamlining theory attributes small cells and genomes to selection for efficient use of nutrients in populations where the population size is large and nutrients limit growth. *Prochlorococcus* and *Pelagibacter* (SAR11) are examples of organisms with huge population sizes but with a small genomes. However, in response to this concern we have replaced the section on this topic with a discussion of the question fast evolution in the CPR.

The Louvain and the hierarchical clustering methods would not in principle cluster them with bacteria that are so divergent.

According to the hhblits manual: “HHsearch and HHblits can detect homologous relationships far beyond the twilight zone, i.e., below 20% sequence identity”. As we performed HMM-HMM comparison using hhblits (as well have re-did the protein clustering), we are confident that our pipeline is enough sensitive to retrieve distant homology.

Other points:

The analysis depends on a number of choices (programs, parameters). These should be tested more thoroughly because they may affect the conclusions.

We agree that the protein clustering will vary with parameters and programs. We report new protein clustering results (different programs, varied parameters) in the revised version and show the main results of the analysis are unchanged.

How important is the default choice of the inflation parameter in MCL ?

Increasing the inflation parameter increases the number of families and results in more protein clusters. After testing the effect of this parameter, we settled on a choice that does not wrongly fragment protein families.

What is the effect of gene annotation in producing genes of heterogeneous sizes and therefore spuriously increase the number of families in the most poorly sequenced genomes (CPR)?

The main way that genome annotation choices impact gene content and gene sizes is if the genomes use an alternative genetic code. This has been well explored for Gracilibacteria (BD1-5) and Absconditabacteria (SR1). Different gene predictors may result in slightly different start positions. We use Prodigal for gene prediction, which is pretty much the industry standard.

What is the effect of using more sensitive homology detection methods?

The revised pipeline is a more sensitive analysis method but its use does not greatly alter the results.

I was intrigued by the claim that type IV pili (tIVp) are very abundant in CPR and rare in nonCPR. There's a lot of data showing that these pili are present in most nonCPR Bacteria and in most Archaea. The claim that they are even more abundant in CPR may result from a clustering problem in fast evolving proteins (the problems mentioned above). Pilins evolve fast in sequence and are short, those of Archaea and Bacteria will not group together by blast (but are homologous as revealed by more sensitive methods). My guess is that the CPR and nonCPR pilins do not cluster together by blastp but would cluster together using more sensitive methods. This is the best explanation for several incoherences in the results: the lack of pilin families in nonCPR in Fig 6 (it is well known that most bacteria encode pilins), the presence of prepilin peptidase in both groups (they are more conserved) which would be unexpected if the nonCPR had fewer pili, and the lack in the list of all the tIVp components that evolve slowly.

We are not saying a capacity enriched in CPR is not present in non-CPR, just that the incidence rates differ. Specifically, T4P are widespread in non-CPR bacteria but they are more abundant in CPR than in other bacteria. With the new clustering, the CPR pilins cluster with the non-CPR bacterial pilins into a single family (fam000005). The manuscript has been revised to reflect these new results.

The section 'functions over-abundant in CPR' is confusing. There are comparisons with functions that seem common in non-CPR Bacteria, leading me to think that this is also a clustering bias and leaving the reader with little information on the extent of the difference between the two groups. To make this clearer one would need to have a precise quantification of the over-abundance of these functions in CPR and nonCPR. I dived into the excel sheet, but that's hard to follow because there are only counts (not controlled for effective sizes nor phylogeny structure).

The main text has been revised to clarify this section. As the reviewer deduced, this information is provided in the annotation table (Table S2). We apologize for the missing information. The revised version of Table S2 includes contingency tables, adjusted p-values and odds-ratio.

The analysis of the correspondence between the clustering analysis and the phylogenetic tree (fig 3C) is very approximative (it's just a succinct verbal argument using the colors matching large fractions of the trees). There are quantitative ways to compare trees that should be used to produce a quantitative (testable) argument.

We agree, and thank the reviewer for this suggestion. We now performed a cophenetic correlation between a 14 ribosomal protein concatenated maximum-likelihood tree and the genome clusterings based on the profiles. The results confirm that the two trees are reasonably well correlated.

L313. There seems to be a misunderstanding on what the literature states on the frequency of competent species. Dubnau and others have indicated that few bacteria are known to be naturally transformable. There is no single CPR known to be transformable if one uses the same criterion (experimental verification). But if one uses the criteria of the paper (a pilus plus some competence genes), then many nonCPR Bacteria would be classed as naturally transformable. So, one should not use Dubnau's paper as a confirmation that these genes are rare in nonCPR against the argument based on the presence of the genes in CPR to state that they are abundant in CPR. The argument should be based on a fair comparison on the frequency of these genes in CPR and nonCPR (and accounting for the fact that these genes may produce distinct protein families In the two groups in the clustering procedure).

The reviewer makes good points. As for the T4P, we do not want to say that competence is widespread in CPR and rare in non-CPR, and now have carefully revised the ms to clarify this. In CPR the comEC genes either a single domain gene or fused to a DUF4131 but their genes do not contain the lactamase_B domain found in some non-CPR genes. This results in two different protein families (one is comEC plus comEC with the lactamase_B fusion, the other is the single domain comEC and proteins with that domain fused to DUF4131). The family enriched in CPR bacteria is the second (lacking the lactamase_B fusion). The manuscript has been modified to clarify this.

I didn't understand the meaning nor the precise details of checking 'annotation admixture'.

Annotation admixture is a way to assess the contamination of the protein families. The rationale is that all members of a family are expected to have the same annotation. For each family with KEGG annotations, we computed the percentage of the KEGG annotations that are different than the most abundant KEGG annotation. For instance, in a family of 8 proteins where 7 members have annotation X and one has annotation Y, the ratio of contamination is 1/8. Most of the families has a ratio of 0, which indicates that the majority of the families have no annotation admixture (Figure S3C). We edited the ms to make it clearer and changed Figure S3 and its legend.

Data is clustered with hierarchical clustering, MCL and Louvain clustering. But it's unclear why the method, or the focus, changes along the text.

We understand that the use of many different clustering methods can be confusing, but we carefully chose each method based on the applications where they are the best according to the literature. We used MCL clustering to create protein families as it is the reference algorithm. The Louvain clustering has been used to detect the modules. This popular algorithm (8581 citations according google scholar) allowed us to keep this block information, which other methods did not. Hierarchical clustering was used to cluster rows (genomes) and columns (families) in the matrices, as it is commonly done for heatmaps.

I found Figures 2A, 3A, 4A, 6 and many in additional online material impossible to read out. These are huge displays of black and white points or lines that are not labelled or highlighted enough to be understood.

We show these matrices as they are interesting and provide an overview of the distribution of the families across the genomes, not just the widespread families on which we focus. The color-codes label the lines (by genome group) and rows (by function). However, it is seemingly impossible to add more detailed labels on these huge diagrams (22,977 col x 2,890 lines). However, the matrix (labelled) as well all the raw data were provided in a .txt format: <https://doi.org/10.6084/m9.figshare.6296987.v1>.

L234. It's a pity that the HMM-HMM analysis is not done on all the families, but only on the predefined set of 6 clusters. Families in small clusters may match other families (probably the case of pilins). This means that the clustering is not necessarily robust regarding clustering parameters (contrary to what it's stated in the text). More general controls are needed to make a convincing case.

We performed HMM-HMM comparison to create all the families, not just the families found in the 6 modules (now 4 modules; see materials and methods, Figure S1). As noted above, in our revised analysis using an improved procedure, the pilins now cluster together.

L482. I found this part of the methods confusing. There is one first clustering by MCL. And then a final clustering by MCL. It is not clear exactly what's the input of this final clustering and how the two differ.

We created a new protein clustering pipeline and simplified the analysis method. We also created a supplementary figure (Figure S1) to help the readers understand the workflow. Our pipeline involves two-step clustering: 1) we used mmseqs2 to create the subfamilies (mmseqs2 performs both the all-vs-all sequences comparison and the first protein sequences clustering using its "set cover" algorithm) 2) we created a HMM for each subfamily multiple alignment and then performed HMM-HMM comparisons. Finally, we performed a MCL clustering to define the families used in the manuscript.

L482. I would suggest making HMM-HMM alignments between families (and not just sub-families) to reveal distant homologies.

Making HMMs require accurate multiple sequence alignments to avoid the creation of spurious HMM models. According to the hhblits manual: “HHsearch and HHblits can detect homologous relationships far beyond the twilight zone, i.e., below 20% sequence identity”. So, our families can contain very divergent protein sequences that do not allow the creation of accurate multiple sequence alignments. We feel that HMM-HMM alignments between subfamilies is sufficient and less prone to spurious alignments. This is the standard procedure, as described in: Bernardes JS, Vieira FRJ, Costa LMM, Zaverucha G. 2015. Evaluation and improvements of clustering algorithms for detecting remote homologous protein families. *BMC Bioinformatics* **16**: 34.

Also, note that hhblits performs an iterative search and takes into account the secondary structure of the protein sequences to improve the sensitivity.

L498. This explanation lacks rigor and quantification.

In this section we explain how we evaluated the number of genomes that have most protein families in each specific module. We did not include results here as this is part of the Methods. The description has been modified to improve clarity.

L477. Which percent identity threshold?

We did not use a percent identity threshold. The percent identities were used as weights in the input network for the MCL clustering. However, with the new pipeline, this clustering has been replaced by the “set cover” clustering provided by mmseqs2.

For the second MCL clustering (HMM-HMM), we also did not use percent identity threshold, but the probability threshold defined by hhblits. We choose the probability score over other metrics because according to the manual: “*The estimated probability of the template to be (at least partly) homologous to your query sequence is the most important criterion to decide whether a template HMM is actually homologous or just a high-scoring chance hit. When it is larger than 95%, say, the homology is nearly certain.*”. We edited the ms to make it clearer.

The list of references has many problems, with volumes, page numbers and sometimes journal names missing. See references 5, 8, 14, 15, 18, 20, 26, 29, 46, 56, etc

Thank you for pointing this out. This has been fixed.

Reviewers' Comments:

Reviewer #1:

Remarks to the Author:

I remain supportive of this study, please find my comments to the revision below.

In general, the authors spend a large part of their available words and figures in support of the argument that CPR is distinct from other bacteria. That might be necessary to convince some critics and be an outcome of peer-review. Personally, I think the final section, that addresses the nature of the differences and dives into the actual biology, is much more interesting! Perhaps Figure 6 could be used to emphasize these findings more in a less abstract and statistical/numerical way.

Minor comments:

P4,5 L104-105: "This discrepancy is due to the fact that the two families partly overlaps to each other according to the HMM-HMM comparison resulting to their splits into two families (Figure S3A)." unclear, please rephrase.

P8, L195 "is are generally", remove "are".

Please explain "single-linkage" and other clustering methods that are discussed in the results – many readers (including this reviewer) will not be able to make sense of the logic without a brief explanation.

To further support the results, could add references to previous phylogenomics studies that have also yielded trees congruent to gene-based phylogenies.

For Figure 4B, should discuss that using the 786 families, the resulting tree is not congruent with gene phylogenies, as many bacterial (super)phyla are not monophyletic in that tree.

P13L307: "This raises the possibility of an alternative systems to remove these toxic ions." => "This raises the possibility of the existence of an alternative system to remove these toxic ions."

P13L312: "The three outer membrane proteins TonB-ExbB-ExbD (fam000056, fam000382 and fam000368) are also missing in CPR. This observation suggests that essentially all CPR lack outer membrane5. Only TonB is an outer membrane protein, ExbBD is responsible for energizing TonB. If you would like to contribute evidence for the absence of an outer membrane, there are many more protein families (e.g. porins, glycolipids, some flagellar genes) that could be engaged to the argument. Alternatively, you might consider rephrasing to " This observation is consistent with the proposed absence of an outer membrane in CPR bacteria5."

P13L315: "Finally, 106 protein families are enriched in CPR and rare in non-CPR bacteria are discussed in detail in the next session." → "Finally, 106 protein families are enriched in CPR, rare in non-CPR bacteria and are discussed in detail below."

Figure 6: I'm wondering if this is the most effective way to visualize these results – also presenting another large matrix like this is not very engaging anymore at this point.

P16L356: "Two non-essential components", non-essential to what?

P16L364: This paragraph, as well as the following ones, contains quite a few minor language

problems. Please revisit text carefully.

P18L412: "we hypothesize they are involved in functions related to the cell envelope." On what basis? Also, the next sentence refers to "observations", which is awkward, as you have just presented a hypothesis, not an observation.

Reviewer #2:

Remarks to the Author:

The authors made a great deal of effort to address my comments/concerns and those raised by the other reviewers, and I thank them for that. With the new clustering pipeline and overall re-analysis, the results obtained seem now more robust to me, plus the new title for the paper better reflects the puzzling identity of the CPR bacteria. However, I still have a few comments.

Major comments:

- What are the "partial CPR genomes" that cluster together on fig. 3? I thought only "high quality" MAGs were used? To me, the >70% completeness criterion used does not correspond to "nearly-complete genomes" (l. 210) that are claimed to have been used. Further, the criteria described in Methods (l. 568): "Genomes with completeness > 70% and contamination < 10% (based on duplicated copies of the SCGs) were considered as near-complete genomes." do not fit either the "high quality" genomes standard according to the MIMAGs paper that the authors co-signed (Bowers et al. 2017, Nat Biotech), but rather to medium to high quality genomes.

Such content-based analyses should only be performed on highly complete genomes... especially if partial ones tend to cluster together. It would be interesting to re-analyse the data with only >90% complete genomes, and/or to map MAGs' completeness along the dendrogram to see if this could explain part of the clustering obtained.

- Fig 1 shows a phylogenetic tree obtained from a previous publication, but if I understood correctly, one was computed for this analysis, which reconstruction is described in Methods (l. 585), but it seems that the tree is not displayed anywhere? It is the tree that was used to compute correlations with clustering on fig S7 for instance, and it should be provided.

- L. 438 in discussion: it is clearly an overstatement to say that the 106 protein families found more in CPR than in non-CPR are "106 essentially CPR-specific genes". Several of them are actually also widespread in CPR. E.g, saying that Type IV pili are CPR-specific is totally wrong. Please rephrase to better reflect the data.

Minor comments:

- Is there a pb with the color legend in Fig S4? I guess red should be grey and vice-versa? Fig S5 inverts the colors from fig S4 between CPR and non-CPR bacteria.

- Fig S6-S7: please write correlation coefficient inside bubbles. Otherwise, levels of correlation are hard to read with the color gradient.

- Fig S11: please for readers' convenience, add "enriched in CPR" and "depleted in CPR" on graphs' titles, and the number of corresponding families in the legend.

- L 307: there's an extra "s" in "systems".
- L. 347-348: please clarify how are they "dependent on externally-derived nucleic acids"?
- L 353-356: A gene fusion could indeed be a shared derived trait of non-CPR VS CPR bacteria. What do the authors mean by "essentially absent" from CPR? Did they find the fused version of the protein in some CPR as well?
- L. 372: upper-case "L" in "GspL".

Reviewer #3:

Remarks to the Author:

The revised manuscript has adequately tackled all the minor comments and the methodological details. For example, the use of mmseqs2 is a good improvement. The claims in the abstract have been tempered down, but I still have problems with them.

- Gene composition follows phylogeny. This is yet presented as an interesting finding (marked in the abstract, mentioned in the results as "Notably," etc). As mentioned before, this is the null neutral model. It's useful information, but it's not remarkable in any way.
- The claim of "novel proteins" in CPR is reasonable. However, one cannot exclude the possibility that many of these are homologous to proteins of the other Bacteria, just too divergent to be identified by sequence similarity methods. For example, the authors now find more homologies between CPR and non-CPR families than previously, because they use more sensitive methods of homology detection. Even HMM-HMM profile alignments can't figure out homology between proteins that are too divergent. It would have been more precise to state that there are many proteins in CPR without recognizable homology to proteins in non-CPR and these may be either very divergent (in which case it is reasonable to suspect that some have acquired novel functions) or actually "novel".
- "The diversity of proteins in CPR may exceed that of other bacteria." 1) The analysis of genes systematically more associated with CPR or non-CPR, shows many more families for the latter (85%) than for the former (15%). 2) Among the families more abundant in relative terms in CPR, those that are actually picked up as examples are for the majority quite abundant in non-CPR too. I understood well the argument that they are even more abundant in CPR, but this is a weak argument for presenting the CPR as a clade with many novel functions, since the ones that are exemplified are also common in non-CPR. 3) The analysis of the four modules of families shows one ubiquitous, two that are mostly absent from CPR, and one that is present in more than 10 CPR phyla. This is not indicative of more diversity of gene families in CPR. 4) I insist that these genomes are necessarily annotated with lower quality than those of non-CPR (not the authors fault, these are drafts from environmental analyses in poorly studied clades: the classic case where sequencing and annotation methods work less well even using state-of-the-art methods). This will increase the number of spurious gene families and these usually don't have homologs. 5) The CPR also probably evolve fast because of their small genome size. I would like to note that the authors have misconstrued my argument about the association between genome size and divergence rates: the argument is valid in general, not just for endosymbionts, see Drake, PNAS, 91 for a classical paper, or Lynch, Nat Rev Gen, 16 for a recent review. Rapid evolution of gene families will lead necessarily - when using these clustering methods - to more gene families than there should be if the methods were able to identify all relations of homology, explaining the effects observed in Fig4. This effect will be as important as the genome reduction is old and as the clade is well separated from the other clades. It should be very important for CPR. Note that a lot of the "specific proteins" are associated with membranes, and these are known to evolve fast. Overall, I could not find any clear evidence for the claim that "The diversity of proteins in CPR may exceed that of other bacteria."

- "The CPR could have arisen in an episode of dramatic but heterogeneous genome reduction or may have arisen from a protogenote community and co-evolved with other bacteria. " The discussion is interesting, but there is actually little novel evidence for distinguish between the scenarios in the paper.

In short, I think this work provides some interesting novel information on this intriguing clade and advances our knowledge on the topic. The results will certainly help this group and others to do further interesting work. But the key conclusions that the CPR and non-CPR are well separated, that CPR have peculiar characteristics and that their genomes are small are not novel, they develop previous knowledge. The conclusion that CPR have more diverse gene families does not seem very convincing. The evolutionary scenarios are interesting, but not clearly distinguished by the data.

Minor issues:

- It should be noted that the association test for CPR vs non-CPR associated gene families assumes that taxa are independent. It does not account for phylogeny. I suspect the more accurate test would decrease the number of significant differences. But it probably wouldn't affect the global picture.

- There are still problems with references: 4, 9 and others lack manuscript identifier, 10 lacks reference to bioarxiv, some journal names are abbreviated, others aren't, etc.

- On the rebuttal: "we are confident that our pipeline is enough sensitive to retrieve distant homology. " This is overly optimistic. As structural biologists know very well, remarkable homology in structure can be impossible to identify in sequence because of excessive sequence divergence. But I agree that the study does the best it can be done at this stage.

Reviewer #1 (Remarks to the Author):

I remain supportive of this study, please find my comments to the revision below.

In general, the authors spend a large part of their available words and figures in support of the argument that CPR is distinct from other bacteria. That might be necessary to convince some critics and be an outcome of peer-review. Personally, I think the final section, that addresses the nature of the differences and dives into the actual biology, is much more interesting! Perhaps Figure 6 could be used to emphasize these findings more in a less abstract and statistical/numerical way.

We thank the reviewer for his comments, and their understanding of the situation

Minor comments:

P4,5 L104-105: "This discrepancy is due to the fact that the two families partly overlaps to each other according to the HMM-HMM comparison resulting to their splits into two families (Figure S3A)." unclear, please rephrase.

We rephrased, it now reads: "Close inspection showed that the two families were not merged because their corresponding HMMs matched only partly (based on the thresholds used, Figure S3A)."

P8, L195 "is are generally", remove "are". - done

Please explain "single-linkage" and other clustering methods that are discussed in the results – many readers (including this reviewer) will not be able to make sense of the logic without a brief explanation.

We added a subsection called "Hierarchical clustering of the genomes and the families" in the Materials and Methods section, it reads:

"The genomes were hierarchically clustered using the Jaccard distance that was calculated based on profiles of protein family presence/absence. The families were also hierarchically clustered based on profiles of presence/absence in genomes. We used an agglomerative (also called bottom-up) method for the hierarchical clustering. In agglomerative clustering, we assign each observation to its own cluster (step 1). Then, compute the similarity (e.g., distance) between each of the clusters (step 2) and then join the two most similar clusters (step 3). Steps 2 and 3 are repeated until there is only a single cluster left. Agglomerative clustering can use various measures to calculate the distance between two clusters. Three different measures were used: single-linkage, complete-linkage and average-linkage. In single-linkage hierarchical clustering, the distance between two clusters is defined as the shortest distance between two points in each cluster. In complete-linkage hierarchical clustering, the distance between two clusters is defined as the longest distance between two points in each cluster. In average-

linkage hierarchical clustering, the distance between two clusters is defined as the average distance between each point in one cluster to every point in the other cluster.”

Reference: https://www.saedsayad.com/clustering_hierarchical.htm

To further support the results, could add references to previous phylogenomics studies that have also yielded trees congruent to gene-based phylogenies.

We cited two papers from Snel *et al.*, 1999 and Chen *et al.*, 2010.

Snel B, Bork P, Huynen MA. 1999. Genome phylogeny based on gene content. *Nat Genet* **21**: 108–110.

Cheng C-H, Yang C-H, Chiu H-T, Lu CL. 2010. Reconstructing genome trees of prokaryotes using overlapping genes. *BMC Bioinformatics* **11**: 102.

For Figure 4B, should discuss that using the 786 families, the resulting tree is not congruent with gene phylogenies, as many bacterial (super)phyla are not monophyletic in that tree.

We discussed it in the ms, it now reads:

“From the hierarchical clustering of the genomes in Figure 4 we generated a tree representation analogous to that in Figure 3B (Figure 4B). Again, the correspondence between genome clusters based on protein family distribution and phylogeny is striking although several phyla are split into several groups. This is particularly apparent for highly diverse phyla such as Actinobacteria, Firmicutes and Proteobacteria (Figure 4B). These incongruences are due to differences in the sets of families (Figure 4A). Interestingly, the groups are not correlated with particular subclades of Actinobacteria, Firmicutes or Proteobacteria. Other phyla, such as Cyanobacteria (bright green in Figure 4A), have more consistent patterns of presence/absence of core protein families. This is reflected in the comparatively short branch lengths in Figure 4B. In contrast, the branch lengths associated with the CPR bacteria are very long.”.

P13L307: “This raises the possibility of an alternative systems to remove these toxic ions.” => “This raises the possibility of the existence of an alternative system to remove these toxic ions.”
- done

P13L312: “The three outer membrane proteins TonB-ExbB-ExbD (fam000056, fam000382 and fam000368) are also missing in CPR. This observation suggests that essentially all CPR lack outer membrane5. Only TonB is an outer membrane protein, ExbBD is responsible for energizing TonB. If you would like to contribute evidence for the absence of an outer membrane, there are many more protein families (e.g. porins, glycolipids, some flagellar genes) that could be engaged to the argument. Alternatively, you might consider rephrasing to “ This observation is consistent with the proposed absence of an outer membrane in CPR bacteria5.”

We thank the reviewer for spotting this error. We have edited the manuscript accordingly, and it now reads: “The three inner membrane proteins TonB-ExbB-ExbD (fam000056,fam000382 and fam000368) are also missing in CPR. The complex interacts with the outer membrane proteins that bind and transport siderophores as well as vitamin B12, nickel chelates, and carbohydrates in Gram-negative Bacteria (Noiraj et al. 2010). This observation is consistent with the proposed absence of an outer membrane in CPR bacteria”.

P13L315: “Finally, 106 protein families are enriched in CPR and rare in non-CPR bacteria are discussed in detail in the next session.” → “Finally, 106 protein families are enriched in CPR, rare in non-CPR bacteria and are discussed in detail below.” - done

Figure 6: I’m wondering if this is the most effective way to visualize these results – also presenting another large matrix like this is not very engaging anymore at this point.

We have carefully considered a variety of ways of presenting the data and find this to be the most effective. We suggest that it is additionally helpful to reuse a presentation format that the reader is familiar with.

P16L356: “Two non-essential components”, non-essential to what?

We edited the manuscript accordingly: “Two other components, ComFC/comFA (fam000096) and ComEA (fam000152), are present, although they are not essential to the natural transformation machinery (Pimentel and Zhang 2018).”.

P16L364: This paragraph, as well as the following ones, contains quite a few minor language problems. Please revisit text carefully.

We carefully revisited the text.

P18L412: “we hypothesize they are involved in functions related to the cell envelope.” On what basis? Also, the next sentence refers to “observations”, which is awkward, as you have just presented a hypothesis, not an observation.

We rephrased, it now reads: “Glycopeptide antibiotics vancomycin inhibit the extracellular steps of bacterial peptidoglycan synthesis. Although the function of vanW is unknown, it has been found in the VanB-type glycopeptide resistance gene cluster in the Gram-positive *Enterococcus faecalis* V583. These observations strengthen the prediction that the cell envelope of CPR bacteria are likely more similar to those of Gram-positive compared to Gram-negative bacteria. The remaining families fam000442, fam000682, fam001505 and fam000479 have no predicted annotations. Given that the three previous families (fam000574, fam000680 and fam000706) are involved in functions related to cell envelope, we hypothesize they are involved in similar functions. “.

Reviewer #2 (Remarks to the Author):

The authors made a great deal of effort to address my comments/concerns and those raised by the other reviewers, and I thank them for that. With the new clustering pipeline and overall re-analysis, the results obtained seem now more robust to me, plus the new title for the paper better reflects the puzzling identity of the CPR bacteria. However, I still have a few comments.

We thank the reviewer for her/his comments.

Major comments:

- What are the “partial CPR genomes” that cluster together on fig. 3? I thought only “high quality” MAGs were used? To me, the >70% completeness criterion used does not correspond to “nearly-complete genomes” (l. 210) that are claimed to have been used. Further, the criteria described in Methods (l. 568): “Genomes with completeness > 70% and contamination < 10% (based on duplicated copies of the SCGs) were considered as near-complete genomes.” do not fit either the “high quality” genomes standard according to the MIMAGs paper that the authors co-signed (Bowers et al. 2017, Nat Biotech), but rather to medium to high quality genomes.

Such content-based analyses should only be performed on highly complete genomes... especially if partial ones tend to cluster together. It would be interesting to re-analyse the data with only >90% complete genomes, and/or to map MAGs’ completeness along the dendrogram to see if this could explain part of the clustering obtained.

Partial CPR genomes are genomes that passed the >70% completeness although they look dubious based on their genome size and after manual curation.

We agree with the reviewer that >70% completeness criterion does not correspond to “near-complete genomes” according to Bowers et al. 2017, Nat Biotech and replaced “near-complete genomes” by “draft-quality genomes”. We also re-analysed the data with only >90% complete genomes (1966 genomes) as suggested by the reviewer and created a supplemental figure similar to Figure 3 (Figure S7). They results are fairly similar to those obtained using the >70% complete genomes (Figure 3). Based on complete-linkage, the cophenetic correlation between a maximum likelihood phylogenetic tree is 0.70, based on average-linkage 0.74, and based on single-linkage, 0.67. Of note, the Dependotiales phylum is still nested within the CPR. We did not re-analyze the data of the NCBI dataset as the genomes are almost all high-quality genomes (2427 out of 2616 genomes have >90 completeness).

We edited the ms, it now reads: “The analysis was made using draft-quality genomes (>70% completeness). It is expected that such content-based analyses is affected by genome completeness. We looked if using high-quality genomes improved the congruence between the phylogenetic tree and the families-content tree. We re-analysed the data using only high-quality genomes (1966 genomes with >90% completeness) (Bowers et al. 2017) (Figure S7). Based on complete-linkage, the cophenetic correlation between a maximum likelihood phylogenetic tree is 0.70, based on average-linkage 0.74, and based on single-linkage, 0.67. Of note, the

Dependentiae phylum is still nested within the CPR (Figure S7). The results are similar to those obtained using the >70% complete genomes (Figure 3).”.

- Fig 1 shows a phylogenetic tree obtained from a previous publication, but if I understood correctly, one was computed for this analysis, which reconstruction is described in Methods (l. 585), but it seems that the tree is not displayed anywhere? It is the tree that was used to compute correlations with clustering on fig S7 for instance, and it should be provided.

We provided the phylogenetic tree to the Figure 3.

- L. 438 in discussion: it is clearly an overstatement to say that the 106 protein families found more in CPR than in non-CPR are “106 essentially CPR-specific genes”. Several of them are actually also widespread in CPR. E.g, saying that Type IV pili are CPR-specific is totally wrong. Please rephrase to better reflect the data.

We rephrased, it now reads: “CPR bacteria separate from other bacteria, including other bacterial symbionts (Figure S15), based on 106 genes that are absent or less abundant in other bacteria.”.

Minor comments:

- Is there a pb with the color legend in Fig S4? I guess red should be grey and vice-versa? Fig S5 inverts the colors from fig S4 between CPR and non-CPR bacteria.

We thank the reviewer for spotting this error, we fixed the legend.

- Fig S6-S7: please write correlation coefficient inside bubbles. Otherwise, levels of correlation are hard to read with the color gradient.

We added the correlation coefficients in Figures S6 and S7.

- Fig S11: please for readers’ convenience, add “enriched in CPR” and “depleted in CPR” on graphs’ titles, and the number of corresponding families in the legend.

We updated the figure S12 with the reviewer’s recommendations.

- L 307: there’s an extra “s” in “systems”. - done

- L. 347-348: please clarify how are they “dependent on externally-derived nucleic acids”?

We clarified it, it now reads: “Given that most CPR lack the ability to *de novo* synthesize nucleotides, it is anticipated that their cells scavenge DNA”.

- L 353-356: *A gene fusion could indeed be a shared derived trait of non-CPR VS CPR bacteria. What do the authors mean by “essentially absent” from CPR? Did they find the fused version of the protein in some CPR as well?*

We found the fusion in 6 CPR genomes, we edited the manuscript, it now reads: “This protein fusion is essentially absent in CPR bacteria (found in six CPR genomes).”.

- L. 372: *upper-case “L” in “GspL”.* - done

Reviewer #3 (Remarks to the Author):

The revised manuscript has adequately tackled all the minor comments and the methodological details. For example, the use of mmseqs2 is a good improvement. The claims in the abstract have been tempered down, but I still have problems with them.

We thank the reviewer for her/his comments.

- Gene composition follows phylogeny. This is yet presented as an interesting finding (marked in the abstract, mentioned in the results as "Notably," etc). As mentioned before, this is the null neutral model. It's useful information, but it's not remarkable in any way.

We agree with the reviewer that it is the null neutral model. We removed "Notably," and added references to two previous phylogenomics studies that have also yielded trees congruent to gene-based phylogenies (Snel *et al.*, 1999 and Chen *et al.*, 2010).

Snel B, Bork P, Huynen MA. 1999. Genome phylogeny based on gene content. *Nat Genet* **21**: 108–110.

Cheng C-H, Yang C-H, Chiu H-T, Lu CL. 2010. Reconstructing genome trees of prokaryotes using overlapping genes. *BMC Bioinformatics* **11**: 102.

- The claim of "novel proteins" in CPR is reasonable. However, one cannot exclude the possibility that many of these are homologous to proteins of the other Bacteria, just too divergent to be identified by sequence similarity methods. For example, the authors now find more homologies between CPR and non-CPR families than previously, because they use more sensitive methods of homology detection. Even HMM-HMM profile alignments can't figure out homology between proteins that are too divergent. It would have been more precise to state that there are many proteins in CPR without recognizable homology to proteins in non-CPR and these may be either very divergent (in which case it is reasonable to suspect that some have acquired novel functions) or actually "novel".

We agree with the reviewer and made a statement in the result section (end of the section Clustering of proteins into families and assessment of cluster quality), it reads: "Although we used sensitive sequence-comparison methods and we assessed the quality of the protein clustering, we cannot completely rule out the possibility that our pipeline failed to retrieve distant homology for highly divergent proteins. Small proteins and fast-evolving proteins are more likely to be affected (Hochstrasser 2009). This lack of sensitivity would result on the separation of homologous proteins into distincts families and would affect the results.". We also replaced "novel proteins" by "proteins without recognizable homology to proteins in other bacteria" in the abstract.

- "The diversity of proteins in CPR may exceed that of other bacteria." 1) The analysis of genes systematically more associated with CPR or non-CPR, shows many more families for the latter (85%) than for the former (15%). 2) Among the families more abundant in relative terms in CPR, those that are actually picked up as examples are for the majority quite abundant in non-CPR too. I understood well the argument that they are even more abundant in CPR, but this is a weak argument for presenting the CPR as a clade with many novel functions, since the ones that are exemplified are also common in non-CPR. 3) The analysis of the four modules of families shows one ubiquitous, two that are mostly absent from CPR, and one that is present in more than 10 CPR phyla. This is not indicative of more diversity of gene families in CPR. 4) I insist that these genomes are necessarily annotated with lower quality than those of non-CPR (not the authors fault, these are drafts from environmental analyses in poorly studied clades: the classic case where sequencing and annotation methods work less well even using state-of-the-art methods). This will increase the number of spurious gene families and these usually don't have homologs. 5) The CPR also probably evolve fast because of their small genome size. I would like to note that the authors have misconstrued my argument about the association between genome size and divergence rates: the argument is valid in general, not just for endosymbionts, see Drake, PNAS, 91 for a classical paper, or Lynch, Nat Rev Gen, 16 for a recent review. Rapid evolution of gene families will lead necessarily - when using these clustering methods - to more gene families than there should be if the methods were able to identify all relations of homology, explaining the effects observed in Fig4. This effect will be as important as the genome reduction is old and as the clade is well separated from the other clades. It should be very important for CPR. Note that a lot of the "specific proteins" are associated with membranes, and these are known to evolve fast. Overall, I could not find any clear evidence for the claim that "The diversity of proteins in CPR may exceed that of other bacteria."

We would like to raise an important point, we do not claim that "The diversity of proteins in CPR may exceed that of other bacteria." but instead "The diversity of combinations of protein families in CPR may exceed that of all other bacteria."

We agree with the reviewer that non-CPR bacteria have more families than CPR. However, we talk about the combination of protein families in CPR not the number of families (1 and 2). This result is reflected by the branch length distribution (boxplots on Figure S9) (3).

We cannot completely rule out sequencing issues, although the diversity measures on Figure S9 have been performed on the diagram obtained with NCBI genomes dataset where 2427 out of 2616 genomes have >90% completeness (4). To explain this diversity, we took into account the fact that CPR are symbionts but we also proposed an alternative hypothesis in the discussion section: "This could be due to genetic drift that resulted in pseudogene formation and gene loss that erased the phylogenetic signal between distant families. Alternatively, the large scale of the CPR may be the consequence of its long evolutionary history. Arguing for the first case, CPR have small genomes and probable symbiotic lifestyles. Thus, they may be analogous to obligate endosymbionts of Eukaryotes, whose reduced genomes are due to genetic drift and small effective population sizes. Counter to this, CPR are not known to be

common endosymbionts and small population sizes for CPR are unlikely, as they are abundant members of microbial communities from diverse environments¹². In the second case, diversity in the core protein family platform may have arisen because symbiotic associations with different groups of bacteria selected for larger or smaller requirements for core biosynthetic capacities in the symbiont.“.

- *"The CPR could have arisen in an episode of dramatic but heterogeneous genome reduction or may have arisen from a protogenote community and co-evolved with other bacteria. " The discussion is interesting, but there is actually little novel evidence for distinguish between the scenarios in the paper.*

We concur. However, we consider the discussion of possibilities worthwhile and interesting!

In short, I think this work provides some interesting novel information on this intriguing clade and advances our knowledge on the topic. The results will certainly help this group and others to do further interesting work. But the key conclusions that the CPR and non-CPR are well separated, that CPR have peculiar characteristics and that their genomes are small are not novel, they develop previous knowledge. The conclusion that CPR have more diverse gene families does not seem very convincing. The evolutionary scenarios are interesting, but not clearly distinguished by the data.

We thank this reviewer for stating that the information is novel and interesting and concur that the results should motivate future interesting work.

Minor issues:

- *It should be noted that the association test for CPR vs non-CPR associated gene families assumes that taxa are independent. It does not account for phylogeny. I suspect the more accurate test would decrease the number of significant differences. But it probably wouldn't affect the global picture.*

We agree with this comment, and defer further analysis to future research.

- *There are still problems with references: 4, 9 and others lack manuscript identifier, 10 lacks reference to bioarxiv, some journal names are abbreviated, others aren't, etc.*

We thank the reviewer for spotting these errors, we fixed them. We used an automatic software to generate the references section, it is unclear if we can use, or not, abbreviated journal names.

- *On the rebuttal: "we are confident that our pipeline is enough sensitive to retrieve distant homology. "This is overly optimistic. As structural biologists know very well, remarkable homology in structure can be impossible to identify in sequence because of excessive sequence divergence. But I agree that the study does the best it can be done at this stage.*

We thank the reviewer, we added a statement regarding distant homology in the result section (see previous comment).